# communications

## psychology

# Reinforcement learning of adaptive control strategies

Leslie K. Held [1✉], Luc Vermeylen[1], David Dignath[2], Wim Notebaert[1], Ruth M. Krebs[1] & Senne Braem [1]

Humans can up- or downregulate the degree to which they rely on task information for goal-directed behaviour, a process often referred to as cognitive control. Adjustments in cognitive control are traditionally studied in response to experienced or expected task-rule conflict. However, recent theories suggest that people can also learn to adapt control settings through reinforcement. Across three preregistered task switching experiments (n = 415), we selectively rewarded correct performance on trials with either more (incongruent) or less (congruent) task-rule conflict. Results confirmed the hypothesis that people rewarded more on incongruent trials showed smaller task-rule congruency effects, thus optimally adapting their control settings to the reward scheme. Using drift diffusion modelling, we further show that this reinforcement of cognitive control may occur through conflict-dependent within-trial adjustments of response thresholds after conflict detection. Together, our findings suggest that, while people remain more efficient at learning stimulus-response associations through reinforcement, they can similarly learn cognitive control strategies through reinforcement.

[1] Department of Experimental Psychology, Ghent University, Henri Dunantlaan 2, 9000 Ghent, Belgium. [2] Department of Psychology, Tübingen University, Schleichstraße 4, 72076 Tübingen, Germany. ✉email: leslie.held@ugent.be

Actions that are shared across tasks are usually easy to implement, while task-specific actions require more reconfiguration. Imagine two tasks, doing the laundry and packing luggage. Some actions are shared between both tasks (folding clothes), while other actions are not (categorizing by fabric only makes sense while doing the laundry, not when packing). Now, when both tasks are carried out in close succession, shared actions will be easier to implement. These necessary reconfigurations show as task-rule congruency effects, i.e., slower and more erroneous responding when dealing with conflicting task sets. It is generally assumed that these reconfigurations indicate that people exert cognitive control. In this study, we investigated whether people can learn adaptive control processes through reinforcement learning.

Traditionally, cognitive control processes, sometimes referred to as executive functions[1], are considered resource-intensive processes that are executed in service of goals. These strategic adjustments are typically considered in reaction to experienced or expected task-rule conflict[2–4]. In connectionist models this conflict arises from shared representations between tasks which create interference[2,5,6]. Thus, control is required to select task-relevant (conflicting) pathways and is traditionally viewed as orthogonal to learning, i.e., the slow building of these pathways and associated weights.

However, recent theories have come to suggest that people may also rely on associative learning or reinforcement learning when it comes to the regulation of these arguably more abstract cognitive processes, i.e., the configurations or weights of control settings themselves[7–13]. This idea is not new. In fact, although the seminal cognitive psychology work of Ulric Neisser similarly referred to these domain-general functions as higher mental processes[14], it also suggested that humans should be able to acquire their own executive routines by learning through experience. Likewise, other early work has shown that people tend to adopt different control strategies in response to different proportions of Stroop incongruent words, as reflected in modulated congruency effects[15,16] (for reviews, see[17,18]). Still, likely due to the dominant focus on these processes as being executive and under strategic control, studies on the (reinforcement) learning of these designated higher control functions are scarce.

It is well established that reward affects control processes (for review see, e.g.[19]). Also more recent studies showed, for example, that blocks of anticipated high vs. low reward can trigger adjustments of control (even in the absence of awareness[20]), that reward prospect improves performance through improvements in task coding[21,22] and that people can be instructed to learn associations between task representations and rewards[23]. However, these studies do not test the idea that cognitive control can be tested through reinforcement learning, i.e., the learning from retrospective rewards.

Therefore, the goal of the current study was to offer a test of the hypothesis that conflict-triggered cognitive control processes are sensitive to reinforcement learning just like stimulus response associations. To focus on learning of higher control functions, we used a design that eliminated learning of more basic stimulus-response associations. Specifically, we ran three pre-registered experiments using a recently developed task switching design that employs unique stimuli on each trial[24,25], and assessed the task-rule congruency effect as a function of selectively rewarding either congruent or incongruent stimuli more. We predicted that selective reinforcement of correct responses to incongruent stimuli should promote control and therefore result in a reduced task-rule congruency effect, i.e., smaller relative differences in reaction times (RT) and accuracy between congruent and incongruent trials, indicated by a significant interaction between reward condition and congruency level. Moreover, we used drift diffusion modelling[26,27] to investigate which underlying cognitive processes were most affected by reinforcement learning.

In all three experiments, participants were asked to categorize target words based on either their size or animacy, depending on a task cue (see Fig. 1). Both tasks used the same response buttons resulting in congruent and incongruent trials. Each experiment had minor differences in design. Most importantly, the third experiment differed from the first two in presenting the task cue and stimulus separated in time, allowing more time for task preparation. Moreover, while all experiments presented each task stimulus only once, Experiment 2 also contained a second experiment half to study the effect of stimulus repetitions. This second half of this experiment was analysed separately and will be referred to as Experiment 2B. In the first part of the paper, we will only present results from Experiments 1, 2, and 3, i.e., where we employed the set of unique stimuli. Importantly, because no stimulus ever re-occurred throughout these experiments and response mappings were orthogonal to the congruency level, rewards were neither contingent on the stimulus, nor the response key, but only on the congruency level.

## Methods

An initial pilot experiment (not reported in the main text) and first two experiments were preregistered on OSF (https://osf.io/qdk5t/) in May, August and December 2021, respectively. These preregistrations slightly differed with respect to the mixed effects models predicting accuracy and RT. The first two preregistrations (for the pilot and first experiment) did not include random stimulus intercepts and the pilot preregistration did not include Block number and Task as a predictor. As we did not detect any effect of time on task on congruency modulations and for better comparison in the final paper, we thus decided to fit a similar model to each experiment excluding Block number. For the combined analysis, we additionally accounted for Experiment number (factorized), as will be described in more detail in the Data analysis section. Complete result tables of all preregistered analyses are displayed in Supplementary Tables 1-6. All experiments were approved by the Ghent University Psychology and Educational Sciences Ethical Committee and participants signed informed consents prior to participation.

**Reporting summary**. Further information on research design is available in the Nature Portfolio Reporting Summary linked to this article.

## Experiment 1

**Participants**. Participants were recruited online from Prolific and compensated for their participation using monetary payment (2.92 pounds). At the end of the experiment, a bonus was determined for each participant and paid out to the 10% highest scoring participants. Sample size was determined based on effect sizes of previous experiments. After pre-processing, our final sample consisted of 104 participants ($N = 68$ women, $N = 2$ undisclosed, $N = 11$ left-handed, Mean age = 24.55 years, SD = 5.14 based on self-report).

**Materials**. The experiment was programmed using JsPsych[28] for online testing. Stimuli consisted of 320 unique English words, adapted from previous studies[24,25]. They consisted of an equal number of words per size/animacy relevant category (i.e., small and animate, small and inanimate, large and animate, large and inanimate). Target words were presented in blue (#0000FF) or green (#00FF00) at the centre of the screen (30px Verdana) on a black background. In 24 practice trials, we included three different feedback messages: CORRECT!, ERROR! And TOO

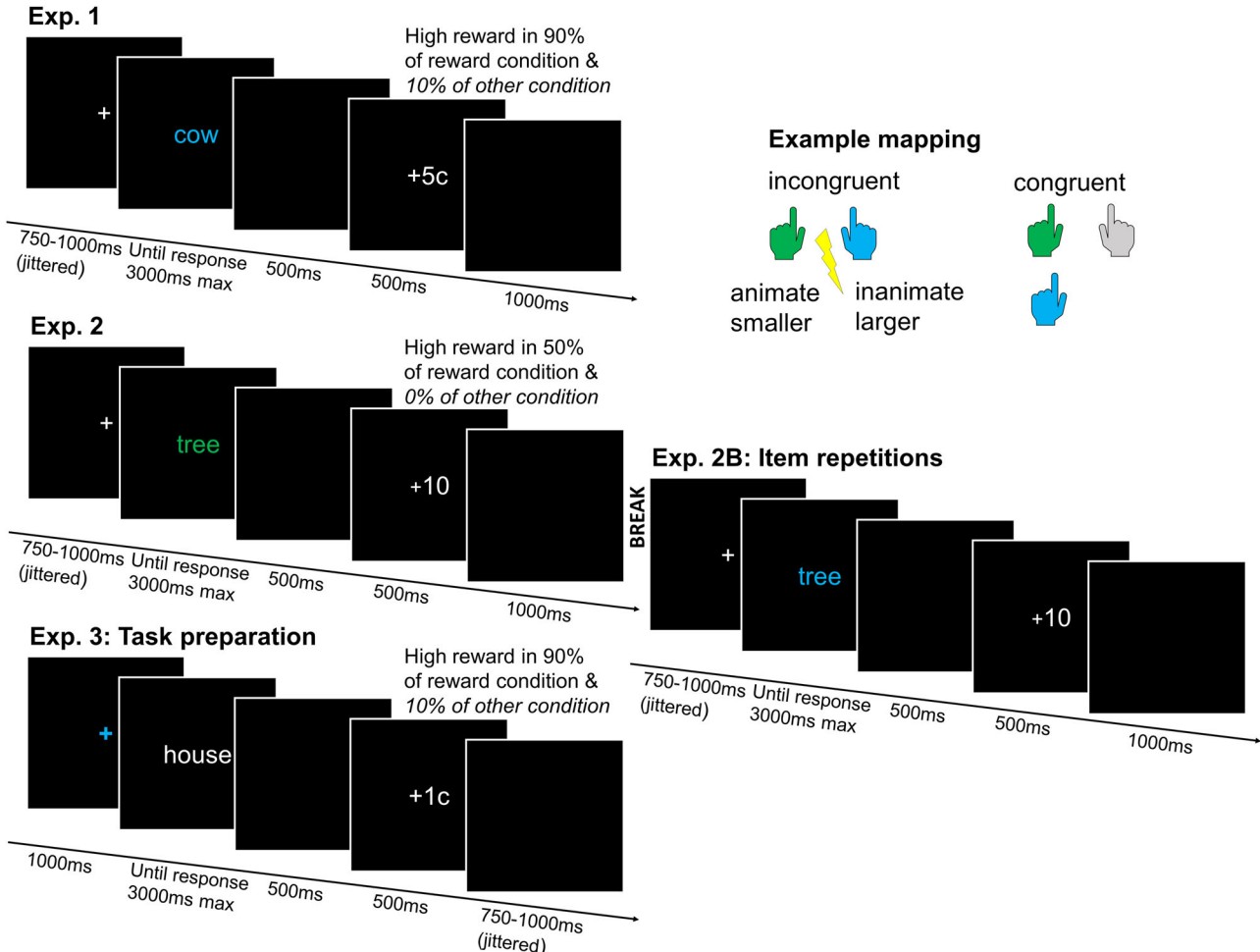

**Fig. 1 Task procedure.** *Note.* General trial procedure for a participant from the group were incongruent trials were rewarded more, with the example task mapping displayed in the right top corner (counter-balanced across participants): congruency was defined by either having to use the same (congruent) or different (incongruent) response button for a given item across both tasks. Rewards were only presented following correct trials, and reward magnitude was dependent on the congruency level of that trial with reward schemes and payout slightly differing across experiments (see Methods). In Experiment 1 and 3, the top 10% of all participants were given the total rewards earned as a bonus payment (+2.92 pounds baseline). In Experiment 2, the top participant of each group was given a 50 Euro gift card (+course credit). Exp.: Experiment.

SLOW! Reward feedback consisted of either a low (1ct) or high (5ct) reward. Task key-to-condition mapping was similar across subjects for the size task (F-key smaller/ J-key larger) and randomized for the animacy task to ensure counter-balancing of task-rule congruency per stimulus (congruent vs. incongruent). Task-rule congruency was defined by stimuli either requiring the same (congruent) or other (incongruent) response as for the currently irrelevant task. For example, "mouse" can represent a congruent stimulus, when "F" refers both to the smaller and animate response key but it can represent an incongruent stimulus in another participant for who "F" refers to the smaller but "J" to the animate response key. We additionally included two questionnaires on reward contingency awareness and the Behavioural inhibition, behavioural activation, and affective responses to impending reward and punishment (BIS/BAS) scale[29]. The latter comprises four subscales measuring BIS, BAS reward responsiveness, BAS drive, and BAS fun seeking with items answered on a four-point Likert scale.

**Procedure**. The total duration of the experiment was around 30–35 min. Participants had to perform two different tasks in four blocks of 80 trials each, following 12 practice trials per task with performance feedback. Depending on the stimulus colour, they either had to perform the size task, in which they had to categorize the stimulus into being smaller or bigger than a basketball, or the animacy task, in which they had to categorize the stimulus into being animate or inanimate. Participants were instructed that animate referred to any kind of organism, including plants, vegetables and fruits and they were encouraged to respond as fast and accurately as possible. If they responded correctly and within 3000 ms, the stimulus was followed by probabilistic reward feedback. The group rewarded more on incongruent trials would receive a high reward following 90% of all incongruent correct trials and 10% of all congruent correct trials, while receiving the low reward on all other correct trials. This mapping was reversed in the group rewarded more on congruent trials. If participants responded incorrectly, they saw a blank screen for the same amount of time as the reward screen. Between blocks, participants obtained feedback on how much (potential) money they earned within that block. At the end of the experiment, they saw the total amount of money they won, and were asked to fill in the questionnaires.

**Statistics and reproducibility**. Our main dependent variables of interest were accuracy and RT. In line with our preregistration, we only registered and analysed data of participants completing

the experiment. On the subject level, we further excluded participants with an overall accuracy of, or below, chance performance (60%) per task. We also excluded participants who showed clear signs of inattentive behaviour, such as long sequences of premature or null responses. For the remaining participants, we excluded missing responses, and trials following a null response or error for all analyses. We also removed trials resulting in an error from the RT analysis. For the RT analyses, we further removed all trials with RTs 1.5 times the inter-quartile range (IQR) above the 75th percentile, and all trials 1.5 times the IQR below the 25th percentile (within-subject) as well as RT faster than 200 ms. Zero sum coding was used for categorical predictors in all analyses. All analyses were conducted based on Bayesian mixed-effects models using brms[30] and RStudio[31]. Our main RT models used shifted lognormal likelihood distributions and the accuracy models used Bernoulli likelihood distributions (logit link). This choice fitted our assumption of a right skewed RT distribution. Our main predictors were Reward group, Congruency, Transition (switch/repeat), as well as all interactions. Task was included as a control variable which did not interact with the other variables. We further used random subject intercepts and random slopes for all within-subject variables and interaction terms. We additionally modelled random intercepts for Stimulus, with Task as a random slope, to control for potential effects of the individual stimuli per task. Default priors were used in both models. The main accuracy model was fitted with 12,000 sampling iterations (3000 warm-up), and the main RT model with 20,000 sampling iterations (10,000 warm-up) each to achieve good convergence (with four chains each).

## Experiment 2

**Participants**. Participants were recruited online from the University's recruitment platform and compensated for their participation with course credit. At the end of the experiment, we determined the highest scoring participant per group and awarded each a 50 Euro gift card. Sample size was doubled due to the non-conclusive effects in the pilot and first experiment. After pre-processing, our final sample consisted of 208 participants ($N = 184$ women, $N = 1$ undisclosed, $N = 21$ left-handed, Mean age = 18.52 years, SD = 1.97). For the analyses on the second half, we had to exclude two more participants, who performed below 60% across all eight experiment blocks (results were virtually identical with or without these participants).

**Materials**. Materials were similar to the first experiment but all stimuli, feedback and questionnaires were presented in Dutch. Reward feedback was no longer monetary but presented as points (1 point or 10 points).

**Procedure**. The total duration of the experiment was around 60 min. The procedure was identical to the first experiment with the crucial difference that in a second experiment half, every stimulus was presented again. In this experiment, the group rewarded more on incongruent trials received the high reward following 50% of all incongruent correct trials and 0% of the congruent correct trials, and the low reward on all other correct trials. This mapping was reversed in the group rewarded more on congruent trials. In addition, we ensured that only one third of the incongruent (congruent) stimuli rewarded with the high reward in the first experimental half was rewarded again with the high reward in the second half. This was done to ensure that high rewards sufficiently varied across stimuli, and participants were not encouraged to develop an item-specific strategy (e.g., when seeing the same items being rewarded a lot).

**Statistical analyses**. Preprocessing was similar to Experiment 1. Participant level exclusion criteria were performed on the first four blocks for the analyses on the first half and across all eight blocks for the analyses on the second half. We further removed the first block of one participant who seemed to have consistently responded with the other response key in one of the two tasks, suggesting misunderstanding of the mapping which was resolved in the later blocks following a reminder. For the analyses on the second half of Experiment 2, we tagged all stimuli that were rewarded with a high reward in the first experiment half for each participant separately, i.e., 50% of the rewarded condition if responded to correctly. Subsequently, for the second half of interest, we created one dataset including only those previously highly rewarded stimuli of the reward condition and all stimuli of the non-rewarded condition. Similarly, we created one dataset containing only stimuli of the reward condition that were not previously rewarded with a high reward and all stimuli of the other condition to test generalization of the reinforcement effect. Analyses on the first experiment half (blocks 1-4) were similar to the ones reported above. Regarding the second half with item repetitions (see above), we ran six models. First, we split the data from the second half according to whether stimuli from the reward condition were paired with a high or low reward in the first experiment half (see above). We then fit the main accuracy and RT models to the set of previously highly rewarded items from the reward condition (and all items from the no reward condition; subset 1) as well as to the set of not previously highly rewarded items from the reward condition (and all items from the no reward condition; subset 2). In addition, to test whether potential effects on previously rewarded stimuli are located at the level of the stimulus or stimulus-response learning, we ran two separate models on subset 1, but further including Task similarity (same vs. different compared to the previous half).

## Experiment 3

**Participants**. Recruitment and payment of participants ($n = 103$ after pre-processing) was similar to Experiment 1. Mean age was 19.11(SD = 4.57), 64 women, 39 men, 11 left-handed.

**Materials**. Materials were similar to Experiment 1 with the only difference that the fixation cross, instead of the target word was now presented in colour.

**Procedure**. The procedure was similar to Experiment 1, however, now the duration of the fixation cross was fixed at 1000 ms and the duration of the blank screen following reward feedback jittered (750–1000 ms).

**Data analysis**. Data analysis and pre-processing was done similarly as in Experiment 1.

## Combined analyses

In order to provide a well-powered test for the global modulation of conflict processing, we generalized across the methodological differences by combining all three experiments into single mixed effects models and a meta-analysis ($N = 415$). Significance was inferred based on the 95% credible interval including 0 or not. In the mixed effects models analysis, we further report the probability of direction (pd), which can be interpreted as a one-sided Bayesian p-value when subtracted from 1 (two-sided when multiplied by 2)[32]. The mixed effects models were similar to the models fitted to each experiment separately, with the exception that we further included experiment number (factorized) as a variable. Exploratory analyses on the BISBAS scores and contingency analyses are displayed in the Supplementary Methods

and Discussion 1 and 2, respectively. Meta-analyses across all four experiments focusing on our main effect of interest, the group by congruency interaction, were run using the package RoBMA[33]. To this end, we generated two different families of models that assume either the presence or absence of an effect, or heterogeneity and combined them using Bayesian model averaging. For the accuracy analysis, where we observed a clear directional effect, we used a normal distribution with mean 0 and a standard deviation of 0.1 for the effect prior treated as belonging to the alternative hypothesis, and a cauchy distribution with the location parameter set to 0 and the scale parameter set to 0.05 for the heterogeneity parameter τ. Both priors were truncated to positive values only as we hypothesized a positive effect and found consistent evidence for this direction. For the meta-analysis on RT, we used a normal distribution with mean 0 and a standard deviation of 0.01 for the effect prior, and a cauchy distribution with the location parameter set to 0 and the scale parameter set to 0.005 for the heterogeneity parameter τ. As we used unstandardized effect sizes, these priors were chosen based on the scale of the effect sizes from the mixed effects models, i.e., assuming a 1% increase or decrease in RT to be meaningful. As we did not observe a clear directional effect across experiments, only the heterogeneity parameter was truncated to positive values only. For both meta-analyses, we used four chains with 2000 warm-up and 5000 sampling iterations.

## Results

Individual estimates of our main effect of interest, i.e., the group by congruency interaction of each sub-model fed in the meta-

analysis is displayed in Fig. 2. Complete result tables for each individual experiment as well as for the joint analyses are presented in Supplementary Tables 7-10. We also ran an initial pilot experiment which, however, showed methodological flaws which is why we excluded it from the main analyses. Notably, our main analyses reached the same statistical conclusions with or without this experiment (also reported in Supplementary Tables 9–12).

**Selective reinforcement of incongruent trials reduces accuracy congruency effects.** In the combined mixed effects model on the accuracy results, we found a significant interaction between Congruency and Group (Est. = 0.038, 95% CI [0.009, 0.067], 99% [pd]). As expected, the group rewarded more on incongruent trials showed a smaller congruency effect than the group rewarded more on congruent trials. Significant main effects were found for Congruency (Est.= 0.299, 95% CI [0.268, 0.330], 100% [pd]), Transition (Est.= 0.139, 95% CI [0.111, 0.166], 100% [pd]), and Task (Est. = 0.244, 95% CI [0.196, 0.292], 100% [pd]), with participants being more accurate on congruent trials, task repetitions, and the animacy task. In line with Goschke[34] we also found a two-way interaction between Congruency and Transition (Est. = −0.029, 95% CI [−0.053, −0.004], 99% [pd]), indicating that congruency effects were smaller following task repetitions. Lastly, we found a significant interaction between Congruency and Experiment (Est. = 0.063, 95% CI [0.019, 0.108], 100% [pd]), suggesting a larger congruency effect in the first experiment compared to the grand mean.

The additionally performed Bayesian model-averaged meta-analysis[35–37] on the estimates of our main effect of interest, the

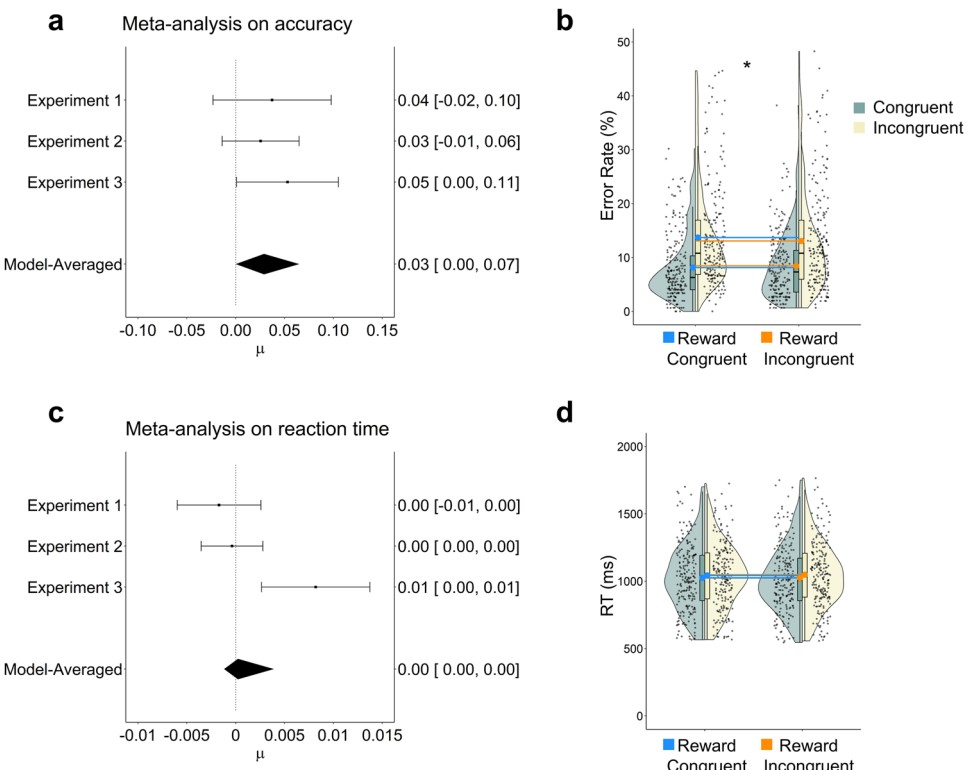

**Fig. 2 Meta-analysis results and congruency effects across experiments** *Note.* **A**, **C** Model estimates per experiment and estimated effect size across experiments for the accuracy (**A**) and RT (**C**) model. The x-axes of the accuracy and RT meta-analyses depict different scaling. μ = Mean effect size. **B**, **D** Raw data plot showing the accuracy and RT congruency effect (interaction between group and congruency). The upper and lower hinges of the boxplot correspond to the first and third quartiles (the 25th and 75th percentiles). The solid points indicate the means in addition to the median. We used blue and orange lines to project the congruency effects on the other group to ease comparison. N = 209 in the condition congruent group, n = 206 in the condition incongruent group. These raincloud plots are adapted from Allen et al.[77]. Significance stars refer to the mixed effects model results, *p < 0.05. Accuracies are depicted as error rates for better visualization.

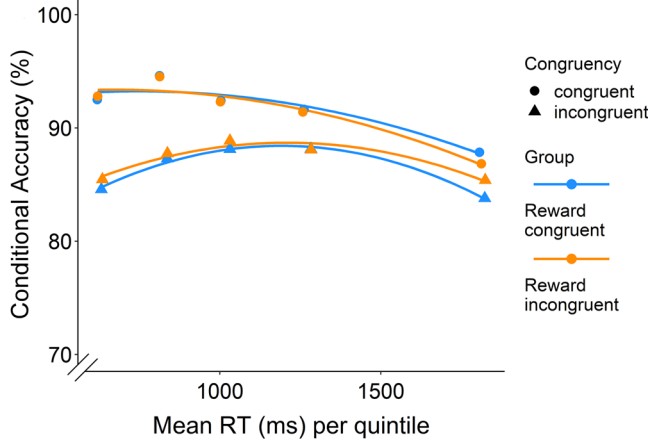

**Fig. 3 Visual explanation of (conflict) drift diffusion models with the boundary shift extension.** *Note.* Drift diffusion and diffusion models of conflict. The green lines refer to congruent trials and the yellow line to incongruent trials (boundary shift for congruent trials not depicted for simplicity).

Group by Congruency interaction across all three experiments revealed the same Group by Congruency interaction across experiments ($\mu = 0.029$, 95% CI [0.000, 0.065], see Fig. 2).

**Selective reinforcement of incongruent trials shows less systematic effects on reaction times.** In our reaction time model, the Group by Congruency interaction did not reach significance (Est. = 0.001, 95% CI [−0.001, 0.004], 86% [pd]). If anything, the effect was in the opposite direction than expected (larger RT congruency effects in the group rewarded more on incongruent trials), suggesting a speed-accuracy trade-off which we will get back to in the following sections. Again, we found significant main effects of Congruency (Est. = −0.015, 95% CI [−0.018, −0.013], 100% [pd]), Transition (Est. = −0.061, 95% CI [−0.065, −0.057], 100% [pd]), and Task (Est. = −0.054, 95% CI [−0.060, −0.048], 100% [pd]), with participants being faster on congruent trials, task repetitions, and the animacy task. We also found a Congruency by Experiment interaction (Est. = −0.006, 95% CI [−0.009, −0.002], 100% [pd]) with congruency effects being larger in the first experiment as compared to the grand mean. Finally, we also observed significant Transition by Experiment interactions for the first (Est. = −0.016, 95% CI [−0.022, −0.010], 100% [pd]) and second (Est. = −0.022, 95% CI [−0.027, −0.017], 100% [pd]) experiment. Inspection of the marginal effects plots and single model fits indicates that switch costs were larger in these first two experiments, where no task cues were presented before target onset and thus less task preparation was possible (in line with e.g.,[38,39]).

In line with the mixed effects model findings, the meta-analysis on RT revealed no clear Group by Congruency interaction but a tendency towards a positive effect, against our initial hypothesis ($\mu = 0.000$, 95% CI [−0.001, 0.004]). This positive effect reached significance in Experiment 3 where task preparation was possible (Est. = 0.007, 95% CI [0.002, 0.013], 100% [pd]).

**Drift diffusion modelling.** In order to better understand the cognitive processes that were affected by our reinforcement schedule and to follow-up on the speed-accuracy trade-off, we turned to drift diffusion modelling (DDM) as an exploratory analysis. DDMs conceptualize binary decision tasks as tasks where the decision maker needs to accumulate evidence until a certain response threshold is reached[26]. They allow us to decompose the decision-making process in at least three parameters: boundary separation (a), capturing the speed-accuracy trade-off or response caution, non-decision time (t), capturing perceptual and motor processing of the stimulus and response, and drift rate (v) capturing the strength of evidence accumulation over time[40] (see Fig. 3). Extensions to the standard DDM, in particular for conflict tasks (diffusion models of conflict; DMC),

**Fig. 4 Conditional accuracy function.** *Note.* Conditional accuracy function. Quintiles were calculated across subjects and conditional accuracy per quintile calculated per congruency and group. A quadratic trendline is added for interpretability. Fast errors can be observed for incongruent trials in both reward groups.

have further included the superimposition of conflicting activations of both controlled and automatic responses by means of modelling evidence accumulation with a gamma function (see Supplementary Methods 3 and[27]). This U-shaped function allows a response on incongruent trials to initially drift towards the boundary of the conflicting response before drifting towards the controlled correct response, thus accounting for fast errors which are not predicted by the standard DDM. The additional parameters comprise at least the peak latency and peak amplitude (describing when this peak occurs and how strong the activation is, see Fig. 3). Another parameter which is often set to a fixed value is alpha describing how much the probability density function shifts to the right. Visual inspection of the conditional accuracy functions (CAF), showing the percentage of correct responding per RT quintile, suggested that a DMC is adequate for our data, as we noted lower accuracy in the fast quintiles on incongruent trials (see Fig. 4).

In order to test if both reward groups differed in one or more of these parameters, we first fitted both the standard DDM and DMC to the data of the three experiments using R and Rcpp[41–43] (See Supplementary Table 13) for used parameter bounds). We fitted separate drift rates for congruent and incongruent trials in the standard DDM (analogous to the DMC) to capture potential differences in evidence accumulation for congruent and incongruent trials. Changes in attentional control are commonly reflected in this parameter[44–46]. For instance, changes in drift rate

have been positively linked to reward incentives[47,48], confidence judgments and reward satisfaction[49], all of which may play an important role in this study. In both models, we further allowed non-decision time to vary trial by trial. Each model was fitted to the data by minimizing the Kolmogorov-Smirnov (KS) criterion, describing the maximum absolute vertical distance between the empirical and the predicted cumulative density functions (CDF) of the response time distributions[50,51] (see Supplementary Table 14 for values). We evaluated the fit by looking at the correlations between estimated and observed quantile means in correct reaction times and estimated and observed accuracy which showed that both models fitted the data well (see Supplementary Methods 4 and Supplementary Table 14, for the according correlations of observed and predicted values and scatter plots). Interestingly, when performing group comparisons on the standard DDM and DMC using $t$-tests, we observed no significant differences between the two reward groups for any of the parameters (see Supplementary Table 15), suggesting we were unable to capture the group difference we observed in accuracy in any of the processes modelled by the traditional DDM and DMC. Thus, these models may not be optimal to describe (changes in) adaptive strategies for conflict processing in our task switching paradigm.

**Dynamic adjustments in decision boundary**. Several studies have suggested that modelling dynamic boundaries which change trial by trial can be helpful in capturing behaviour. For example, it has been shown that "collapsing" boundaries can help in modelling some urgency signal[52–54] or, alternatively, that people strategically adjust their decision boundary in response to conflict on a trial-by-trial basis[55–58]. Similarly, we recently reasoned that people, across a range of decision making tasks from cognitive conflict to moral or reasoning conflict tasks, may dynamically and strategically adjust their boundary within a trial, i.e., after the detection of conflict[59]. Perhaps our participants used such strategic adjustments of decision boundary to differentially respond to congruent versus incongruent trials, depending on which congruency condition was reinforced, which also fits with the observation that the group difference in congruency effect was most pronounced for the slowest reaction times (Fig. 4). In order to test whether people can learn to strategically adjust their boundary as a function of differentially reinforced congruency conditions, the model has to allow for an additional boundary shift parameter (which could be positive or negative) after a shift time (independent from but after the non-decision time). While several researchers have hypothesized about these congruency-dependent, within-trial shifts in decision boundary, we are currently unaware of published efforts to actually model and estimate them. Therefore, we extended the DDM and DMC by estimating versions where the decision boundary was allowed to change (both increase or decrease) given a certain shift point in time. We estimated these individual shift time parameters, which were allowed to vary trial by trial by adding normally distributed noise. Both models seemed to fit our data well, although the conflict DMC with boundary shifts fitted the data better compared to the standard DDM with boundary shifts (see Supplementary Methods 4 and Supplementary Table 14) for the according correlations of observed and predicted values and scatter plots. Both models with boundary shifts had lower KS values than those without.

Interestingly, allowing boundary shifts resulted in group differences in the differential boundary shifts for congruent versus incongruent trials, reaching marginal significance in the best fitting DMC with shifts (Mean difference score for boundary

adjustments congruent group=11.42, SD = 29.35, Mean difference score incongruent group = 6.11, SD = 27.89; $t(413) = 1.89$, $p = .06$, Cohen's d = 0.19, 95% CI [−0.01, 0.38], see Fig. 5 and Fig. 6), and significance in the standard DDM with shifts (Mean difference score congruent group=0.01, SD = 0.03, Mean difference score incongruent group=0.00, SD = 0.03; $t(413) = 3.01$, $p = .003$, Cohen's d = 0.30, 95% CI [0.10, 0.49]). We also followed up on these interaction effects with between-group comparisons for congruent and incongruent trials separately, which suggested that this effect could not be attributed to either of the congruency conditions in the DMC with shifts, but suggested higher boundary shifts for incongruent trials, $t(399.54) = −2.29$, $p = .02$, Cohen's d = −0.23, 95% CI [−0.42−−0.03] (Welch correction to degrees of freedom due to inequality of variances). We found no significant effect for congruent trials, $t(413) = 0.26$, $p = .79$, Cohen's d = 0.03, 95% CI [−0.17−0.22], in the group rewarded more for incongruent trials in the standard DDM with shifts. Specifically, as can be seen in Fig. 5, our modelling results suggest that, while both groups upregulated their decision boundaries more in response to congruent trials, the group rewarded more on incongruent trials also upregulated their decision boundary on incongruent trials (see Discussion for interpretation). While a higher boundary for congruent trials might seem at odds with the general observation of congruency effects in RT, it is important to note that the RT is also determined by the drift rate, which was generally higher (i.e., faster) for congruent than incongruent trials. The DDM with shifts further revealed group differences in the difference score in drift rate, suggesting a smaller difference in drift rate for congruent versus incongruent trials when incongruent trials were rewarded more (Mean congruent group = 0.03, SD = 0.05, Mean incongruent group = 0.02, SD = 0.05), $t(413) = 2.43$, $p =.02$, Cohen's d = 0.24, 95% CI [0.04–0.43]). In follow-up analyses there was a marginally significant effect for reward group on the drift rate in incongruent trials, $t(403.85) = −1.66$, $p = .098$, Cohen's d = −0.16, 95% CI [−0.36–0.03]. There was no statistically significant effect for the reward group in congruent trials, $t(413) = 0.25$, $p = .81$, Cohen's d = 0.02, 95% CI [−0.17–0.22]. None of the other parameters were able to explain the difference in congruency processing (see Fig. 5 and Fig. 6). The finding of numerically similar adaptations in drift rate fits with current theories on cognitive control, as reinforcement or incentives have been linked to faster and more accurate responding[48,60,61].

To see if either differential adjustments to boundary shifts or drift rate on congruent or incongruent trials were adaptive in terms of optimizing performance, we also conducted additional optimality analyses. As we only modelled separate drift rates for each trial type in the DDM with shifts, these optimality analyses were restricted to this model. Specifically, we simulated 50000 trials for 256 agents with different combinations for each boundary shift (−0.05-0.1 in 16 steps of 0.01) and drift rate (0-0.5 in 16 steps of 0.033) while setting all other parameters to average estimates across both groups. Reward was discounted by dividing it through the reaction time plus the intertrial interval, to account for the rewarding effect of fast responses (based on[44]). Interestingly, these simulations showed that, indeed, it is beneficial to upregulate one's boundary shift in response to the reward condition up to a certain point after which it ceases to be beneficial. Moreover, it is adaptive to increase one's drift rate for the reward condition (see Fig. 7). While increasing one's boundary after a certain point ceases to be beneficial as the time cost increases without an important benefit for additional reward, it is plausible that the drift rate is constrained by biological factors, such as attentional processing units.

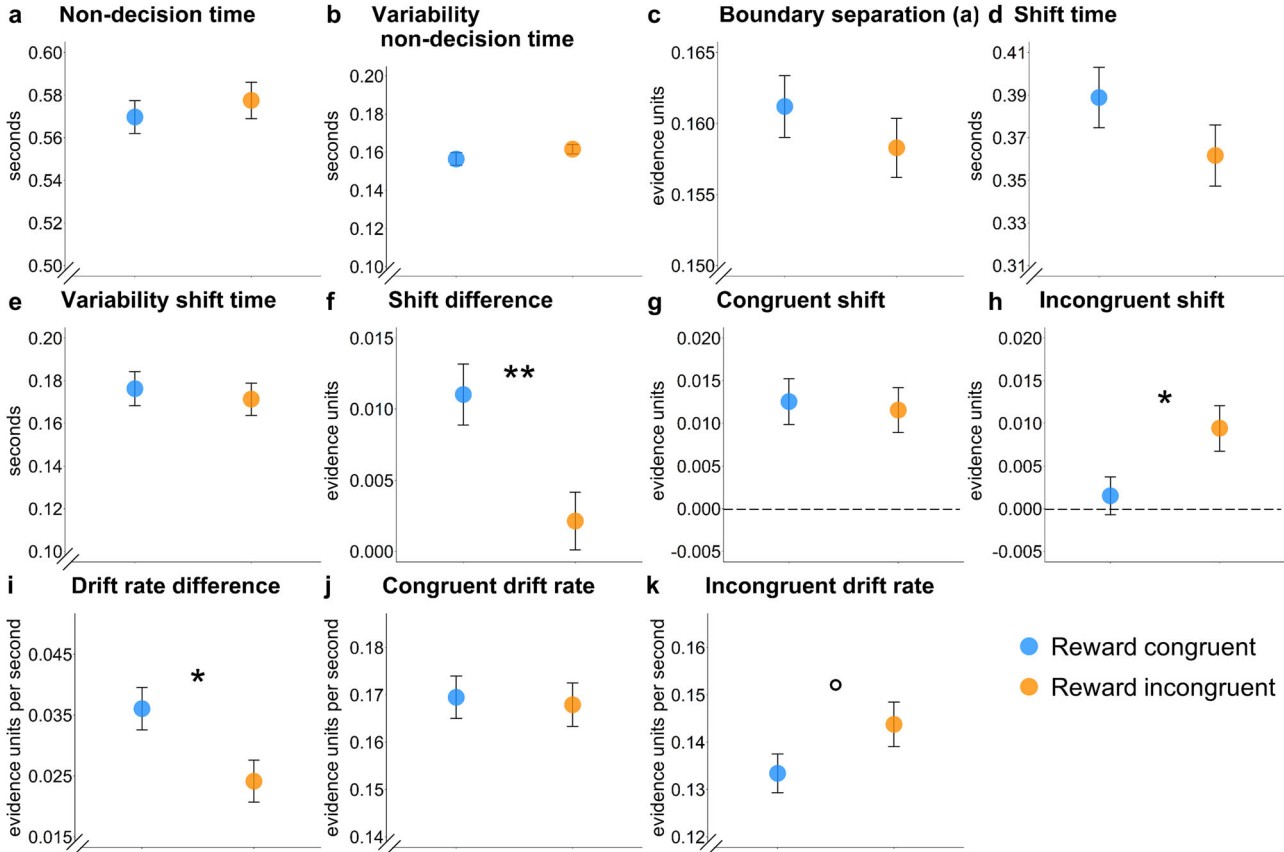

**Fig. 5 Modelling results of the standard drift diffusion model (DDM) with boundary shifts.** *Note.* Estimates of the different model parameters per group (**a–k**). Bars reflect the standard error per group. The group rewarded more on incongruent trials seemed to increase their boundary more for incongruent trials than the group rewarded more on congruent trials. In fitting the models, we went with the common formalizations of the classic DDM, modelling boundary separation as the difference between boundaries. **\*\****p* < 0.01, **\****p* < 0.05, °*p* < 0.1.

**Allowing selective reinforcement learning of stimuli.** In all previous sections, we tested a design without stimulus repetitions to ensure that the selective reinforcement of the different congruency conditions could not be attributed to the selective reinforcement of a specific stimulus or a specific response. We reasoned that in a design with stimulus repetitions, the repeated reinforcement of, e.g., the word "elephant" could give rise to differential processing rather than the mere congruency identity of the stimulus. To put this hypothesis to test and compare the magnitude of our global reinforcement effect with an item-specific one, we added an additional experiment half to Experiment 2, where we repeated every stimulus of the first four blocks once. Crucially, our reward schedule in this experiment allowed us to distinguish whether recurring stimuli of each respective reward condition were presented with the high reward or the low reward in the first experiment half (see Methods section). Thus, we could test the difference of item-specific (previously highly rewarded item of reward condition) vs. congruency specific (global; not previously highly rewarded item of reward condition) reinforcement effects (see Method section for details). We found that the Group by Congruency interaction reached significance in the accuracy model in the set of stimuli containing the previously highly rewarded items of the reward condition (and all items of the other condition; Est. = 0.103, 95% CI [0.054, 0.153], 100% [pd]), but not in the set of stimuli containing the not previously highly rewarded items of the reward condition (and all items of the other condition, Est. = 0.026, 95% CI [−0.024, 0.076], 85% [pd]). Again, the group rewarded more on incongruent trials showed smaller congruency effects than the group rewarded more on congruent trials (see Fig. 8). This effect was of a markedly larger effect size in the previously rewarded items as compared to the findings of the first experiment half discussed previously, while the (non-significant) effect-size of the not previously rewarded items corresponded to the effect size found in the first half and our across-experiment analysis. Interestingly, we also found this interaction to be significant in the expected direction in the RT model in the set of stimuli rewarded previously with the high reward (Est. = −0.004, 95% CI [−0.007, −0.001], 99% [pd]) but not in the stimuli not previously rewarded with the high reward (Est. = −0.002, 95% CI [−0.006, 0.002], 80% [pd]; see Fig. 8 for comparison). In other words, we found a speeded processing of incongruent stimuli when they were rewarded more, rather than the seemingly slower processing reported in the main analyses across experiments without stimulus repetitions. Perhaps, items previously highly rewarded now, when recognized as such, act as reward cues themselves, a possibility that did not exist in the design with no item repetitions. In line with our finding, such reward cues have been associated with invigorating effects that speed up (rather than slow down) responses[60,61]. It is important to note that stimuli rewarded with the high reward in the first experiment half, could be presented with both the same or the other task in the second half and thus with the same or different response in the reward incongruent group. In order to test if response repetitions played a role, we additionally ran the model including the similarity of task as a factor. These additional control analyses revealed no significant three-way interaction between group, congruency and similarity of tasks, suggesting a reinforcement effect at the stimulus rather than the stimulus-

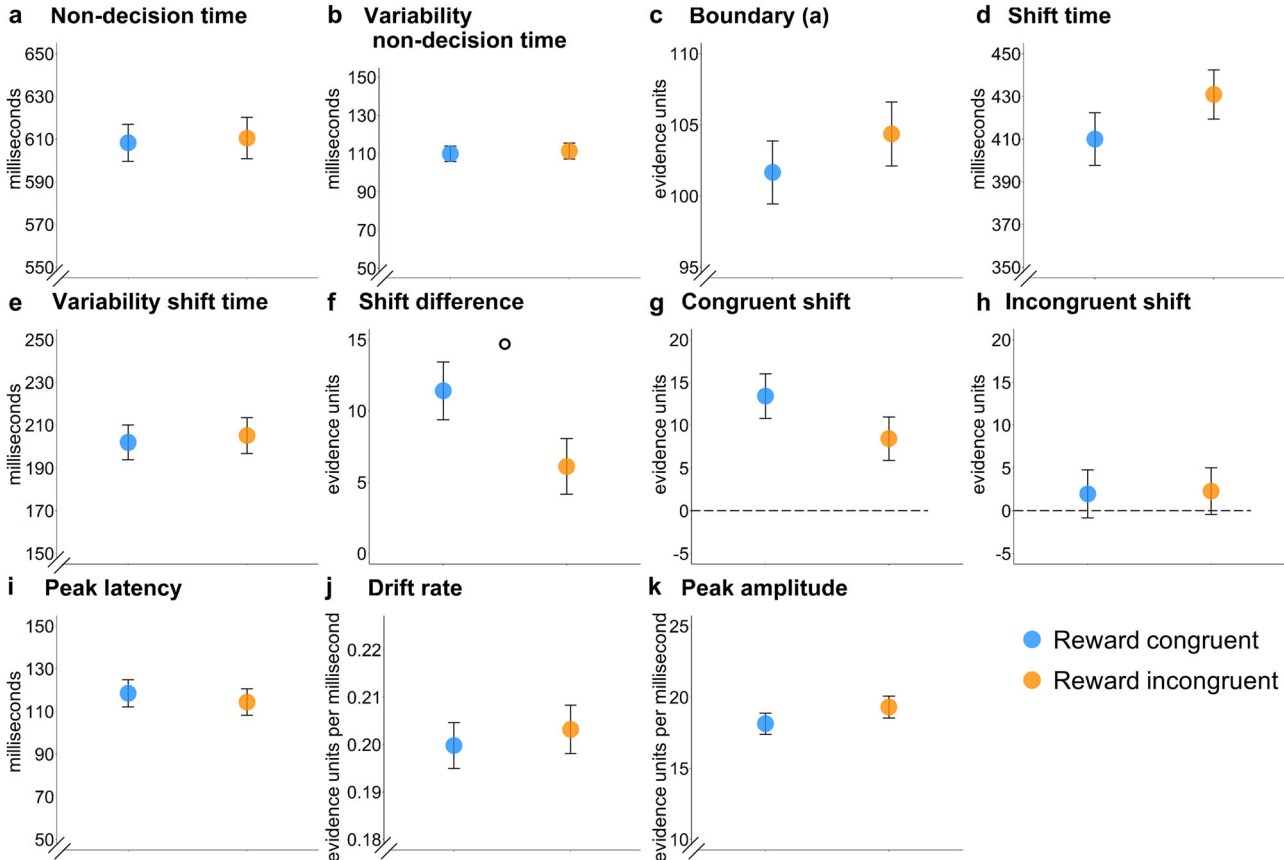

**Fig. 6 Modelling results of the diffusion model of conflict (DMC) with boundary shifts** *Note.* Estimates of the different model parameters per group (**a–k**). Bars reflect the standard error per group. In fitting the models, we went with the common formalizations of the DMC, modelling boundary as the relative difference of the lower and upper bound from the start of the evidence accumulation (here from 0). °$p < 0.1$.

response level (see Supplementary Tables 16–19) for results tables). Interestingly, we also found a significant Group by Transition interaction in the accuracy models, both in the set of previously highly rewarded (Est. = −0.047, 95% CI [−0.092, −0.002], 98% [pd]), and not highly rewarded stimuli (Est. = −0.061, 95% CI [−0.103, −0.019], 100% [pd]), with the group rewarded more on incongruent trials showing larger switch costs than the group rewarded more on congruent trials. These results align with attractor state models, conceptualizing states of cognitive flexibility and stability as landscapes with wells that differ in terms of depth[6,62]. If the group rewarded more on incongruent trials shows deeper wells, therefore shielding their task representations better, this may come at the cost of larger switch costs required to overcome this shielding. The previously found main effects of task, transition and congruency remained significant in both accuracy and RT models (corresponding to the earlier described effects; all pd= 100%).

## Discussion

Across three experiments, we tested whether reward can teach people abstract control processes. A joint analysis showed that people indeed learned to increase their control settings when incongruent trials were rewarded more. Computational models suggest that this effect can be attributed to a selective upregulation of decision boundaries. It has been known for long that reinforcers shape learning of stimulus-response links[63]. Indeed, more recent studies suggested that this could also explain selective reinforcement of congruency effects in Stroop- and Simon- like tasks[64–67]. However, the present results critically extend this line of research by showing that reinforcement learning is not restricted to describe acquisition of concrete stimulus-response associations, but can also successfully capture learning of abstract control processes independently of the concrete S-R links they operate on (see also[65,66]).

In a similar vein, we found that stimulus-specific reinforcement learning (when using stimulus repetitions in Experiment 2) resulted in noticeably stronger modulations of the congruency effect in accuracy, and qualitatively different patterns in reaction time, relative to the reinforcement learning of adaptive control strategies (when using unique stimuli; as in the combined analysis of Experiments 1, 3, and the first half of Experiment 2). Together, these findings suggest that reinforcement learning can operate on different processes that vary along a gradient of abstraction, where the learning of stimulus value likely is a more efficient strategy for control selection, followed by increasingly abstract forms of learning when the former is not possible[68–70]. It would be interesting for future research to formalize this idea as a credit assignment problem where policies at different levels of control can have different learning rates. Specifically, such studies could test whether participants are biased towards first assigning rewards to the stimulus-response level rather than their more domain-general control configurations, and whether this first level is also associated with higher learning rates.

It is important to note that additional cognitive processes may have contributed to the observed group differences in conflict processing. For instance, rewards, rather than having served as a reinforcement signal in the narrowest interpretation, may have acted as a form of conscious reminders to pay more attention to the task, particularly when following incongruent stimuli. Yet, while offering a complementary explanation of our findings, this

## Optimal boundary shift per group and congruency level

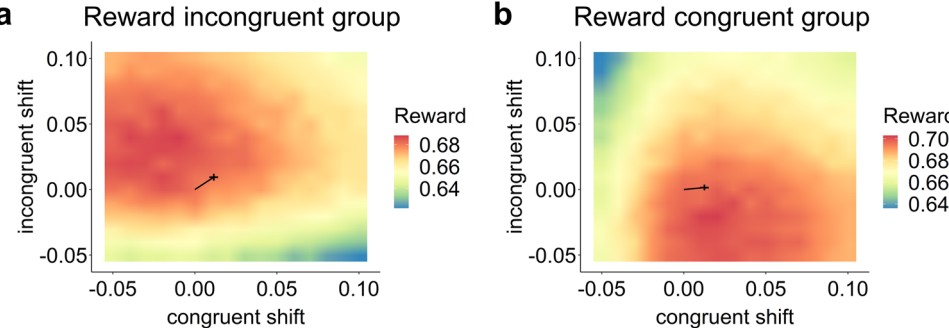

**Fig. 7 Heat maps of analyses for drift diffusion model with boundary shift.** *Note.* Figure depicting mean discounted reward rates based on different combinations of boundary shifts (in number of evidence units; **a**, **b**) and drift rates (in number of evidence units per second; **c**, **d**) for each congruency condition per reward group. Scaling is set to five breaking points for rewards but differs between sub plots. Crosses depict estimated means and standard errors across participant estimates. The line segment for the optimal boundary shift plots depicts the difference to the initial boundary with coordinates (0, 0).

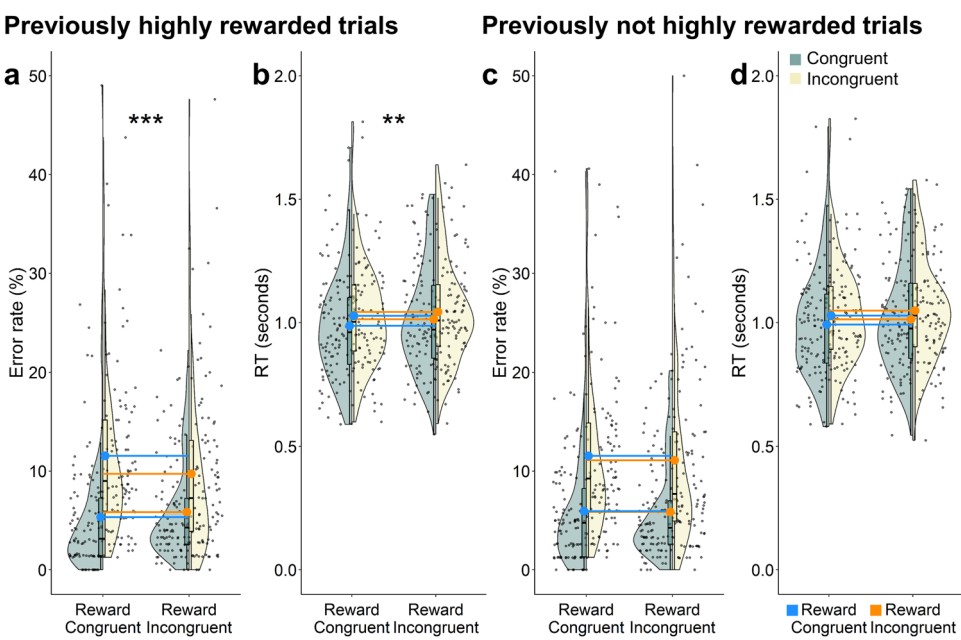

**Fig. 8 Stimulus-specific and stimulus-unspecific reward effects in the second experiment half.** *Note.* Raw data plot showing the accuracy (**a**, **c**) and RT (**b**, **d**) congruency effect (interaction between group and congruency) of the previously highly rewarded trials (**a**, **b**) and previously not highly rewarded trials (**c**, **d**). The upper and lower hinges of the boxplot correspond to the first and third quartiles (the 25th and 75th percentiles). The solid connected points indicate the means in addition to the median. We used blue and orange lines to project the congruency effects on the other group to ease comparison. $N = 104$ in the condition congruent group, $n = 102$ in the condition incongruent group. ***$p < 0.001$, **$p < 0.01$.

account cannot stand alone or inherently rule out a reinforcement learning account, because, if observed differences in control modulations were the mere consequence of participants' redirecting their attention, we would expect the resulting lasting control modulations to be global rather than congruency-specific. As elaborated in the Introduction, we believe that an encompassing learning perspective, in both cases, provides a more parsimonious explanation of cognitive control adaptations rather than conceptualizing learning and control as strictly orthogonal processes. To further investigate what underlying cognitive processes might drive the found effect of reinforcement learning, we ran four different drift diffusion models by factorially combining the decomposition of the drift rate into a controlled and automatic process to model the early interference (making it a DMC[27]), and the implementation of a congruency-dependent shift in decision boundary after the detection of stimulus (in)congruency. In other words, we compared a standard DDM with a DDM with varying decision boundary, with a DMC, and with a DMC with varying decision boundary. The latter was inspired by previous studies showing that participants show higher decision thresholds, i.e., cautiousness when conflict increases[55–58], allowing more time and caution for conflict resolution upon conflict detection. The idea of boundary (or speed-accuracy)[71] adjustments as a reward maximizing strategy has further gained attention in the design of neural networks, providing a plausible mechanism to be tested in actual performance[13].

Interestingly, while both the standard and diffusion model of conflict revealed no group differences in parameters, the shift in boundary parameter turned out to be critical in explaining the group differences we observed in accuracy. While both groups increased their boundaries following congruent trials, the group rewarded more on incongruent trials also showed a similar increase for incongruent trials. The general increase for congruent trials across groups may seem surprising at first, as we would have expected a larger increase in response to the more difficult trials. However, the observed pattern may be a result of the specific reinforcement schedule, and is compatible with previous research showing that people are only willing to invest effort if it pays off[72]. Assuming that congruent trials require less effort or control to resolve, people may have been willing to invest more time on these trials (i.e., by being more cautious at the cost of foregoing speed), for both high or low rewards. Doing the same for incongruent trials, however, might have been considered more costly, and therefore only resulted in increased decision boundaries in the group that was also rewarded more on incongruent trials, in line with the basic premise behind the expected value of control theory[73]. Naturally, this interpretation should be taken with caution as it is derived post-hoc on the basis of exploratory follow-up analyses.

Following a similar pattern as compared to the boundary shift, the group rewarded more on incongruent trials showed a smaller difference in drift rate for congruent and incongruent trials according to the DDM with shift, pointing at control adaptations not only in response caution but also to smaller differences in selective attention or improved task focus between the two congruency conditions. This finding shows that control can be optimized in multivariate ways[45], as also further supported by our optimality analyses. Assuming that participants' behaviour was driven by a reward maximization strategy, i.e., the maximization of rewards per time unit, both adjustments in drift rate as well as in boundary shift were adaptive strategies up to a certain point, after which boundary shifts ceased to be optimal.

In addition to the above described findings across experiments, we note that there were a few experiment-specific findings. One of the most important between-experiment differences, perhaps, was that we presented the target and task simultaneously in our first two experiments, allowing participants less time to process and respond to the target, while in the last experiment, the task cue preceded the target. Initially, we reasoned that we could maximize variance by presenting task and target information simultaneously, providing more leeway for a potential effect of our congruency-specific reinforcement schedule: if participants knew only upon target presentation which task to perform, the competing task information can conflict more with the relevant task. However, in hindsight, this design likely worked against the reinforcement-sensitive regulation of congruency-specific control strategies, which turned out to be most outspoken in our third experiment where the task cue and target were temporally separated. In this experiment, participants could prepare first for the task at hand (as evidenced by the reduced switch costs), and focus more on task- and congruency-specific control strategies during target processing and response preparation.

**Limitations**. To follow up on our main results, i.e., the reduction of congruency effects as a function of reinforcement learning, we used diffusion models as described above. These models were of exploratory nature to obtain better insights into which underlying cognitive mechanisms were affected by our reinforcement schedule. However, these models were not preregistered, and do not model the learning of these process parameters. Therefore, it would be interesting for future research to follow up on our findings by developing a more overarching computational model that can account for both congruency-specific reinforcement learning and control adaptations (linked to performance benefits), for example by combining an evidence accumulation module to a neural network, and assigning separate weights to congruency-specific boundary shift parameters that can become stronger with practice. Along those lines, it would be interesting to adapt neural networks such as by Simen and colleagues[13] to capture adaptations in thresholds as a function of stimulus congruency.

## Conclusion

In sum, we provide evidence that congruency-specific control strategies can be subject to reinforcement learning, extending previous studies that focused on the reinforcement of stimulus, response, and task values. Through modelling, we show that people most likely optimized control by strategically shifting their decision boundary in a congruency-specific manner. At the same time, our findings also suggest that reinforcement learning at the lowest available level (i.e., increasing stimulus value) is still the preferred, default strategy guiding control adaptations, resulting in stronger and faster changes in performance. This fits with the broader ideas that reinforcement learning of abstract control strategies can only follow the slower process of abstraction (i.e., requiring exposure to multiple stimuli) and meta-learning[74], and that the strategies on which it operates are considered more costly[72,75].

## Data availability

All data and materials are available on OSF under this link: https://osf.io/qdk5t/[76].

## Code availability

All analysis code to program the experiment and run the analyses is available on OSF under this link: https://osf.io/qdk5t/[76].

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

## Acknowledgements

This work was supported by an ERC Starting grant awarded to Senne Braem. (European Union's Horizon 2020 research and innovation program, Grant agreement 852570), and a fellowship by the Fonds Voor Wetenschappelijk Onderzoek–Research Foundation Flanders 11C2322N to Leslie K. Held and 11H5619N to Luc Vermeylen The funders had no role in study design, data collection and analysis, decision to publish or preparation of the manuscript.

## Authors contributions

Leslie K. Held: Led conception and design, acquisition of data, analysis and interpretation of the data, drafted and revised the article and agreed the submitted version for publication. Luc Vermeylen: Contributed to conception and design, acquisition of data, analysis and interpretation of the data, revised the article and agreed the submitted version for publication. David Dignath: Contributed to conception and design, revised the article and agreed the submitted version for publication. Wim Notebaert: Contributed to conception and design, revised the article and agreed the submitted version for publication. Ruth M. Krebs: Contributed to conception and design, revised the article and agreed the submitted version for publication. Senne Braem: Contributed to conception and design, acquisition of data, interpretation of the data, revised the article and agreed the submitted version for publication.

## Competing interest

The authors declare no competing interests.
