## [Peer Review File · Communications Psychology]

23rd Jun 23

Dear Ms Held,

Thank you for your patience during the peer-review process. Your manuscript titled "Reinforcement learning of adaptive control strategies" has now been seen by 3 reviewers, whose comments are appended below. You will see that they find your work of some potential interest. However, they have raised quite substantial concerns that must be addressed. In light of these comments, we cannot accept the manuscript for publication, but would be interested in considering a revised version that fully addresses these serious concerns.

We hope you will find the Reviewers' comments useful as you decide how to proceed. Should additional work allow you to address these criticisms, we would be happy to look at a substantially revised manuscript. If you choose to take up this option, please highlight all changes in the manuscript text file, and provide a detailed point-by-point reply to the reviewers.

The reviewers collectively highlight that the rationale, methods and interpretation of the results are insufficiently clear. Please address all their concerns and questions with care. This will require not only presentational changes to improve clarity and reporting, but also additional analysis and a justification of modelling assumptions.

You will also need to revise the manuscript to bring it into alignment with our statistics reporting and interpretation guidelines. You can find more on this in the attached template and checklist (see below); in brief, statistics must be reported in full wherever they appear, and in particular summary statements, such as "all $p > x$, all $t < z$ " are not permitted. Marginally significant results may be reported, but not discussed. Non-significant findings in NHST may not be taken as evidence for the absence of an effect or difference and therefore may not be interpreted. Any such claims must be supported by appropriate positive evidence for the null, which can be derived from Bayesian statistics or equivalence tests. There is no statistical test that can demonstrate absence of an effect. Statements such as 'There is no difference between x and y .' or 'X does not affect Y.' must be revised to read 'We found [no/little] credible evidence of a difference between x and y .' or 'We found [no/little] credible evidence that X affects Y.'

Please work with the template and checklist to inform formatting of text and sections as you address the reviewer concerns regarding structure and clarity of the manuscript.

All exploratory analysis needs to be labelled as such.

If the revision process takes significantly longer than five months, we will be happy to reconsider your paper at a later date, provided it still presents a significant contribution to the literature at that stage.

Please use the following link to submit your revised manuscript, point-by-point response to the Reviewers' comments with a list of your changes to the manuscript text (which should be in a separate document to any cover letter) and any completed checklist:

[link redacted]

Please do not hesitate to contact me if you have any questions or would like to discuss the required revisions further. Thank you for the opportunity to review your work.

Best regards,

Antonia Eisenkoeck

Antonia Eisenkoeck
Senior Editor
Communications Psychology

EDITORIAL POLICIES AND FORMATTING

Editorial Policy: [Policy requirements](https://www.nature.com/documents/nr-editorial-policy-checklist.pdf) (Download the link to your computer as a PDF.)

Furthermore, please align your manuscript with our format requirements, which are summarized on the following checklist:

[Communications Psychology formatting checklist](https://www.nature.com/documents/commspsychol-style-formatting-checklist-article-rr.pdf)

and also in our style and formatting guide [Communications Psychology formatting guide](https://www.nature.com/documents/commspsychol-style-formatting-guide-accept.pdf) .

* TRANSPARENT PEER REVIEW: Communications Psychology uses a transparent peer review system. This means that we publish the editorial decision letters including Reviewers' comments to the

authors and the author rebuttal letters online as a supplementary peer review file. However, on author request, confidential information and data can be removed from the published reviewer reports and rebuttal letters prior to publication. If your manuscript has been previously reviewed at another journal, those Reviewers' comments would not form part of the published peer review file.

* **CODE AVAILABILITY:** All Communications Psychology manuscripts must include a section titled "Code Availability" at the end of the methods section. In the event of publication, we require that the custom analysis code supporting your conclusions is made available in a publicly accessible repository; please choose a repository that provides a DOI for the code; the link to the repository and the DOI must be included in the Code Availability statement. Publication as Supplementary Information will not suffice. We ask you to prepare and upload code at this stage, to avoid delays later on in the process.

* **DATA AVAILABILITY:**

All Communications Psychology research manuscripts must include a section titled "Data Availability" at the end of the Methods section or main text (if no Methods). More information on this policy, is available at <http://www.nature.com/authors/policies/data/data-availability-statements-data-citations.pdf>.

At a minimum the Data availability statement must explain how the data can be obtained and whether there are any restrictions on data sharing. Communications Psychology strongly endorses open sharing of data. If you do make your data openly available, please include in the statement:

We recommend submitting the data to discipline-specific, community-recognized repositories, where possible and a list of recommended repositories is provided at <http://www.nature.com/sdata/policies/repositories>.

If a community resource is unavailable, data can be submitted to generalist repositories such as [figshare](https://figshare.com/) or [Dryad Digital Repository](http://datadryad.org/). Please provide a unique identifier for the data (for example a DOI or a permanent URL) in the data availability statement, if possible. If the repository does not provide identifiers, we encourage authors to supply the search terms that will return the data. For data that have been obtained from publicly available sources, please provide a URL and the specific data product name in the data availability statement. Data with a DOI should be further cited in the methods reference section.

REVIEWER EXPERTISE:

Reviewer #1: cognitive control

Reviewer #2: cognitive control, drift diffusion modelling

Reviewer #3: reinforcement learning

Reviewer #1 (Remarks to the Author):

I had the opportunity to review the manuscript "Reinforcement learning of adaptive control strategies" that has been submitted for publication in Communications Psychology. Therein the authors present results from three preregistered task switching experiments (plus an additional pilot experiment) that provide first evidence for reinforcement learning of abstract, "higher level" cognitive control adaptations. When performance in incongruent trials is selectively more rewarded than in congruent trials, the congruency effect was reduced in the accuracy data. To explore this effect further, the authors developed and applied a new kind of drift diffusion model, and tested the influence of reinforcement of the stimulus in one of the experiments.

While I find this research very innovative, the methods sound, and the results new and interesting, I am not sure about the fit of this research to Communications Psychology. My major concern is not about the quality of the research, but rather to the fact that the format seems not appropriate. Three experiments (plus a pilot experiment), a newly developed version of the drift diffusion model, and an additional experiment half of a single experiment are reported in very condensed form, which made it very challenging for me to follow and often created the wish for more elaboration on many occasions. A lot of relevant and interesting information is only briefly mentioned in the main text or only provided in one of the 8 (!) appendices, which further complicated the read. It took me several times of reading to really understand everything. Therefore, in my opinion, a more traditional article format would be a better fit to this research. I recommend to introduce and report each experiment individually first, followed by the overarching analyses. In addition, I would like to see a more elaborate introduction and general discussion.

Here are some concrete instances where I found the text to be too condensed or insufficient and some additional minor points (in chronological order):

- p. 3, first paragraph: "Actions that are shared across tasks are usually easy to implement, while task-specific actions require more reconfiguration" -> for me, a spelled-out example would have been helpful to ease understanding of this sentence
- Introduction (in general): an overview of the present study at the end of the introduction would have been nice
- p. 4/5, first line: "... providing a relatively higher reward in 90 (10)% of the reward (other) condition, we rewarded 50 (0)% of the reward (other) condition with the relatively higher reward in the second experiment" -> I found it very hard to unpack the meaning of this sentence
- p. 8, Figure 2: I would add to the note that both meta-analyses have a different scaling on the x-axis. Furthermore, I would like to see a graph depicting not only difference scores, but the full design of this study, so that one can see whether the reduction in the congruency effect is due to congruent trials, incongruent trials, or both. Maybe, at least as another appendix ;-)
- p. 8 ff., Drift diffusion modelling: it should be explicitly mentioned that these are not preregistered, but exploratory analyses; furthermore, I was wondering if there was a reason to conduct both, the standard DDM and the DMC? And why are the methods on the diffusion modelling reported here and not in the methods section?
- p. 14, Figure 5: on the left side of this figure, blue and orange are used to differentiate congruent

and incongruent trials, while on the right side blue and orange are used to differentiate the two groups. I suggest to use a distinct set of colors for each factor. The note seems to explain only the left side of the figure without mentioning the panels on the right side.

- p. 16, second half: "... with the group rewarded more on incongruent trials showing larger switch costs than the group rewarded more on incongruent trials" -> the second "incongruent" should read "congruent"

- pp. 17-20: the discussion feels overly short with respect to the number of analyses and results reported in this manuscript. To give one concrete example, I would have liked to see a discussion not only of the decision boundary, but also the drift rate effect.

- p. 21 ff., Methods: I found it hard to follow due to the simultaneous report of the three different experiments.

- p. 31, Appendix A: please consider a revision of Table A2. When reading the content of the table, I was very confused about the Frequency Low-High part, because there was no mentioning of congruency. I guess it makes sense if one takes the table note into account. But it is again very complicated to unpack the relevant information. I hope there is a more accessible way to provide this information.

Reviewer #2 (Remarks to the Author):

The question addressed by the manuscript is whether cognitive control can be adjusted by reinforcement learning via reward contingent on correct responses for congruent vs. incongruent stimuli, respectively. 3 experiments are reported supporting this contingency and the effect is concluded (based on DDM-modelling) to be mediated by dynamic (and stimulus specific) adjustment in the response boundary.

I believe that the studies are well executed and that they address important questions. However, there are few major weakness that I estimate to reduce the impact this paper will receive: i) a lack of theoretical clarity in the rationale of the study and in its motivation, ii) a tendency to oversell (and over-interpret) the results, iii) the adequacy and clarity of the computational modeling, iv) a relatively unclear/ineffective writing that doesn't help the reader to understand the logic and the motivation of the study or its conclusion. Below I will try to explain each of these points with some examples.

1. Rationale and motivation

As the authors state in their introduction, the research literature considers learning (reinforcement or supervised) and cognitive control as fundamentally different processes. This sets the stage to their present attempt to show that this distinction overlooks the ability to use reinforcement to adjust control.

IN a way, such a result is not novel, as in the early 1990s, Tzelgov et al (1992) and others, have shown that a mere manipulation of the frequency of neutral-words can modulate the magnitude of Stroop interference:

Tzelgov, J., Henik, A., & Berger, J. (1992). Controlling Stroop effects by manipulating expectations for color words. *Memory & Cognition*, 20(6), 727–735. <https://doi.org/10.3758/BF03202722>

What the present study has that differs from the Tzelgov et al (and many others), is that here the

type of stimuli does not vary in the amount of conflict, but the critical variable is the feedback that is contingent on stimulus congruency. It is not clear to me however, that this difference has anything to do with reinforcement learning. I will try to explain my lack of clarity below (which I expect to be shared by other readers). There are two main points:

i) There is a good reason why learning is typically seen as orthogonal to control. As it was clearly explained in the classical model of Stroop (Cohen, Dunbar, McClelland, 1990), the problem of control is to overcome the effect of learning (which would result in a strong but irrelevant pathway dominating a weaker but relevant pathway). Unlike learning which takes multiple trials (i.e., practice), control can take place upon instruction (from a first trial), and allows us to respond in a task-relevant way (with little practice).

ii) The authors distinguish between the learning of stimulus specific associations, and the learning of congruency, presenting their study as a case of the second. However, it is not clear to me that this situation has anything to do with learning in the sense of reinforcement (or supervised) learning algorithms. This is because learning algorithms are mediated by changes in synaptic connections and this study does little to show how a neural-network that carries out their task may use the feedback to make changes on the connections of the network to account for the outcomes. Instead, it seems easier to conceptualize of the congruency reward as a reminder given to subjects to pay attention to relevant task. In the case of congruent trials the reminder is not very effective (because both tasks point to the same response), however for incongruent stimuli the reward succeeds to remind the subjects that they should keep paying attention to the relevant task. I would thus expect that the results would be very similar, if the congruency based reward would be replaced (after each trial or after every few trials) by a cue reminding the participants to pay attention to the relevant task.

If this was the case, I don't think we would consider this to have anything to do with reinforcement learning, but rather to show that control can be adjusted by conscious reminders. To conclude, without showing that the change in control can be obtained as a result of a learning algorithm based on the neural network that executes the task and is presented with the feedback that the subjects received, it is hard to really accept the motivation for considering this study to show anything qualitatively different from the previous demonstrations that control processes can be adapted (as in Tzelgov et al.)

2) Over-interpretation of the results

To show a change in control (or selective attention) one needs to show that the subject is able to change focus from the irrelevant (but automatic task or dimension) to the relevant one. Such a change would typically correspond to consistent changes in both speed and accuracy (and not, as in this study, in a speed accuracy tradeoff). In other words, it should result in a change of the drift value (and not in the chance of the response boundary). In the later case (which is what the modelling seems to confirm) we have a modulation of the caution that subjects apply to some type of stimuli, but not an enhanced focus on the task-relevant dimension.

I could see that there may be a weaker sense of labeling such a change of stimulus specific caution as a (weaker) type of control, but this at least requires a clear discussion of the distinction as well as qualification of the results.

3) The adequacy and clarity of the computational modeling

For me a main weakness of the computational modeling is that it didn't rely on some accepted/standard computational modeling of conflict (such as, for example, Botvinick et al, 2001). I can understand applying DDM modeling as an attempt to simplify the task. However, I am less familiar with the conflict version of the DDM, and I expect that many readers may be unfamiliar with it too.

Thus if one wishes to rely on it (rather than do the proper job of applying a mechanistic conflict model), one should, at least, explain this model in much more detail to the audience. As it reads now, there appear to be quite a number of unmotivated and unclear assumptions:

i) why does the boundary shifts upwards? Most choice-RT studies seem to report, either fixed boundaries or boundaries that shift downwards (indicating a type of urgency)!

ii) What is the functional form of the drift variable for the incongruent condition?

iii) The various variables associated with this conflict-DDM model need to be better explained...(peak-latency, peak-amplitude). At present those appear somewhat unclear and arbitrary.

iv) Unclear/ineffective writing

I will give few examples (I will add questions/comments, IN CAPITALS) to explain difficulties to understand:

p. 3 (1st paragraph of the Intro):

To engage in goal-directed behaviour, we often rely on higher-order task representations. Actions that are shared across tasks are usually easy to implement, while task-specific actions require more reconfiguration. These necessary reconfigurations show as task-rule congruency effects, i.e., slower and more erroneous responding when dealing with conflicting task sets. It is generally assumed that these reconfigurations indicate that people exert "cognitive control". In this study, we investigated whether people can learn adaptive control processes through reinforcement learning.

IT IS NOT CLEAR WHAT IS MEANT BY HIGHER ORDER REPRESENTATIONS (HIGHER COMPARED WITH WHAT?), NEITHER IT IS CLEAR WHAT THIS HAS TO DO WITH RECONFIGURATIONS AND WITH CONGRUENCY EFFECTS. PERHAPS AN EXAMPLE COULD HELP. I believe that the first paragraph needs to be as clear as possible (and free from jargon) to attract readers to keep reading.

p. 3 (2nd paragraph).

Recent theories have come to suggest that people may also rely on associative learning or reinforcement learning when it comes to the regulation of these arguably more abstract cognitive processes (e.g., Abrahamse et al., 2016; Braem & Egner, 2018; Chiu & Egner, 2019; Doebel, 2020; Logan, 1988; Verbruggen et al., 2014). This idea is not new. In fact, although the seminal cognitive psychology work of Ulric Neisser similarly referred to these functions as "higher" mental processes (Neisser, 1967), it also suggested that humans should be able to acquire their own executive routines by learning through experience. Still, likely due to the dominant focus on these

processes as being executive and under strategic control, studies on the (reinforcement) learning of these designated "higher" control functions are scarce.

THE TERMS MORE ABSTRACT AND HIGHER LEVEL ARE NOT CLEAR. WHAT EXACTLY IS ABSTRACT (IN EXECUTING, SAY, A STROOP TASK)? AND WHY IS HIGH LEVEL? (COMPARED WITH WHAT?)

I SUPPOSE THAT WHAT THE AUTHORS INTEND HERE IS TO DISCUSS THE DISTINCTION BETWEEN LEARNING PROCEESS AND ATTENTIONAL CONTROL, BUT PLEASE SEE COMMENT ON THE RATIONALE & MOTIVATION, ABOVE...

p. 16

Interestingly, we also found a significant Group by Transition interaction in the accuracy models, both in the set of previously highly rewarded (Est = -0.047, 95% CI [-0.092, -0.002], 98% [pd]), and not highly rewarded stimuli (Est = -0.061, 95% CI [-0.103, -0.019], 100% [pd]), with the group rewarded more on incongruent trials showing larger switch costs than the group rewarded more on congruent trials.

group rewarded more on INCONGRUENT trials showing larger switch costs than the group rewarded more on CONGRUENT trials.

THERE MUST BE SOME MISTAKE HERE...

In addition, I find that some of the Figures are not very clear.

For example, in Fig 5 (which is possibly the most important figure), the random trajectories are hard to see, so I suggest to eliminate them, so that one can focus on the model parameters.

It may help also to present not only the difference in parameters between the congruent and incongruent stimuli, but each of them separately. Also what is meant by "boundary separation"? separation between the two response boundaries or between the same response boundary in different conditions/groups?

Finally, I found the mixed Method (All experiments grouped together) somewhat difficult to read, and making the understanding of the task as well as the motivation for the various experiments more difficult.

Reviewer #3 (Remarks to the Author):

The authors present data that suggests adaptive control can be trained through reinforcement. Across three task-switching experiments, one group was rewarded highly for correct responses on congruent trials, and another group was rewarded highly for correct responses on incongruent trials.

Congruency effects (faster/more accurate responding for congruent than incongruent trials) in accuracy were reduced when incongruent trials were highly reinforced. Interestingly, this group showed a larger congruency effect on RT, suggesting a speed-accuracy trade off. However, this was not able to be captured in drift diffusion modelling. Using modified drift diffusion models, the authors instead suggest this is due to an increase in boundaries for choice, reflecting greater caution on high reward incongruent trials.

The paper is nicely written, and the data and modeling would be of interest to readers. I have a few comments.

It would be useful to share model code in OSF, particularly if this is the first effort to model and estimate shifts in decision boundaries.

A recent relevant paper perhaps worth citing is Diao, L., Li, W., Chang, W., & Ma, Q. (2022). Reward Modulates Unconsciously Triggered Adaptive Control Processes. *i-Perception*, 13(1).

<https://doi.org/10.1177/20416695211073819>

Also potentially relevant:

Wisniewski D., Reverberi C., Momennejad I., Kahnt T., Haynes J. D. (2015). The role of the parietal Cortex in the representation of task-reward associations. *Journal of Neuroscience*, 35(36), 12355–12365. <https://doi.org/10.1523/Jneurosci.4882-14.2015>

Hall-McMaster S., Muhle-Karbe P. S., Myers N. E., Stokes M. G. (2019). Reward boosts neural coding of task rules to optimise cognitive flexibility. *Journal of Neuroscience*, 39(43), 8549–8561.

<https://doi.org/10.1523/JNEUROSCI.0631-19.2019>

Etzel J. A., Cole M. W., Zacks J. M., Kay K. N., Braver T. S. (2015). Reward motivation enhances task coding in frontoparietal Cortex. *Cerebral Cortex*, 26(4), 1647–1659.

<https://doi.org/10.1093/cercor/bhu327>

Regarding the increase in boundary in high reward trials – the interpretation that the group rewarded more on incongruent trials increased their decision boundary more for incongruent trials than the group rewarded more on congruent trials is clear. The result that both groups upregulate decision boundaries on congruent trials is indeed surprising, and initially seems at odds with the congruency effect in RT. The discussion addresses how this can be interpreted for the group differences observed in accuracy, but can the authors make clearer whether these models and this interpretation can account for the congruency effects in RT and the unexpected greater congruency effect for the rewarded incongruent group? From my interpretation of Figure 5, it seems like the DDM with shift can but the DMC with shift cannot, is this correct? I should note that I have limited experience with drift diffusion models, but I imagine some readers will also, so some additional clarity here may be helpful.

For the data using previously reinforced stimuli, the authors note “we found a speeded processing of incongruent stimuli when they were rewarded more, rather than the seemingly slower processing reported in the main analyses across experiments without stimulus repetitions. This suggests that previously highly rewarded stimuli now acted as reward cues, which have been associated with invigorating effects that speed up responses.” – Could you explain why this type of reward is more likely to speed up responses than the reward in the experiments without stimulus repetitions?

P18. “It would be interesting for future research to formalize this idea as a credit assignment

problem where policies at different levels of control can have different learning rates.” – I didn’t understand what this meant.

Ghent, 25 October, 2023

Dear Reviewers,

Thank you for reviewing our manuscript (*COMMSPSYCHOL-23-0112*), entitled “Reinforcement learning of adaptive control strategies”. We were pleased to read that you find our work interesting, and made changes to the manuscript to address your comments which we believe improved it a lot. Specifically, we now changed the format, presenting each experiment separately in the methods section in order to reach a better fit for the journal. We further formulated some of our claims more carefully and expanded our modelling section by adding optimality analyses, additional plots as well as a more elaborate discussion and interpretation of the drift rate parameter. We provide a point-by-point reply to all comments below, detailing our response to each concern separately.

Sincerely,

Leslie Held,

also on behalf of Luc Vermeulen, David Dignath, Wim Notebaert, Ruth Krebs and Senne Braem

Reply to Reviewer #1

[...]

While I find this research very innovative, the methods sound, and the results new and interesting, I am not sure about the fit of this research to Communications Psychology. My major concern is not about the quality of the research, but rather to the fact that the format seems not appropriate. Three experiments (plus a pilot experiment), a newly developed version of the drift diffusion model, and an additional experiment half of a single experiment are reported in very condensed form, which made it very challenging for me to follow and often created the wish for more elaboration on many occasions. A lot of relevant and interesting information is only briefly mentioned in the main text or only provided in one of the 8 (!) appendices, which further complicated the read. It took me several times of reading to really understand everything. Therefore, in my opinion, a more traditional article format would be a better fit to this research. I recommend to introduce and report each experiment individually first, followed by the overarching analyses. In addition, I would like to see a more elaborate introduction and general discussion.

We agree that the previous submission was very condensed. Following your suggestion, we now adopted a more traditional format of the article and introduce each experiment separately in the methods section and refer more explicitly to the main result per sub-model in the meta-analysis Figure (see response below). We also added information that is helpful for understanding our design to the main text. Finally, we added more information to the design Figure. We believe that the current version is therefore better suited for *Communications Psychology*. More detailed responses are given point-by-point below.

Here are some concrete instances where I found the text to be too condensed or insufficient and some additional minor points (in chronological order):

- p. 3, first paragraph: "Actions that are shared across tasks are usually easy to implement, while task-specific actions require more reconfiguration" -> for me, a spelled-out example would have been helpful to ease understanding of this sentence

We thank you for this suggestion. We now added an example to the beginning of our introduction, p. 3, l. 38:

Actions that are shared across tasks are usually easy to implement, while task-specific actions require more reconfiguration. Imagine two tasks, doing the laundry and packing luggage. Some actions are shared between both tasks (folding clothes), while other actions are not (categorising by fabric only makes sense while doing the laundry, not when packing). Now, when both tasks are carried out in close succession, shared actions will be easier to implement.

- Introduction (in general): an overview of the present study at the end of the introduction would have been nice

In line with this suggestion, we now moved the introductory paragraph, previously labelled as “Task description” from the beginning of the Results section to the end of the Introduction and further modified it to (1) highlight main differences between the different experiments and (2) provide a rough outline of our paper, p. 5, l. 87:

In all three experiments, participants were asked to categorize target words based on either their size or animacy, depending on a task cue (see Figure 1). Both tasks used the same response buttons resulting in congruent and incongruent trials. Each experiment had minor differences in design. Most importantly, the third experiment differed from the first two in presenting the task cue and stimulus separated in time, allowing more time for task preparation. Moreover, while all experiments presented each task stimulus only once, Experiment 2 also contained a second experiment half to study the effect of stimulus repetitions. This second half of this experiment was analysed separately and will be referred to as Experiment 2B. In the first part of the paper, we will only present results from Experiment 1, and 3, i.e., where we employed the set of unique stimuli. Importantly, because no stimulus ever re-occurred throughout these experiments and response-mappings were orthogonal to the congruency level, rewards were neither contingent on the stimulus, nor the response key, but only on the congruency level.

We have also added more detail to the overview of the design in Figure 1.

Figure 1

Task procedure

Note. General trial procedure for a participant from the group were incongruent trials were rewarded more, with the example task mapping displayed in the right top corner (counter-balanced across participants): congruency was defined by either having to use the same (congruent) or different (incongruent) response button for a given item across both tasks. Rewards were only presented following correct trials, and reward magnitude was dependent on the congruency level of that trial with reward schemes and payout slightly differing across experiments (see Methods). In Experiment 1 and 3, the top 10% of all participants were given the total rewards earned as a bonus payment (+ 2.92 pounds baseline). In Experiment 2, the top participant of each group was given a 50 Euro gift card (+ course credit). Exp.: Experiment.

And we highlighted where the relevant main results per experiment can be found (in the meta-analysis Figure), p. 5, l. 107:

Individual estimates of our main effect of interest, i.e., the group by congruency interaction of each sub-model fed in the meta-analysis is displayed in Figure 2.

Complete result tables for each individual experiment as well as for the joint analyses are presented in S1-3.

- p. 4/5, first line: “... providing a relatively higher reward in 90 (10)% of the reward (other) condition, we rewarded 50 (0)% of the reward (other) condition with the relatively higher reward in the second experiment” -> I found it very hard to unpack the meaning of this sentence

We thank you for pointing this out. We now unpacked this sentence for each experiment separately,

On p. 21, l. 468:

The group rewarded more on incongruent trials would receive a high reward following 90% of all incongruent correct trials and 10% of all congruent correct trials, while receiving the low reward on all other correct trials. This mapping was reversed in the group rewarded more on congruent trials.

And on p. 23, l. 520:

In this experiment [Experiment 2], the group rewarded more on incongruent trials received the high reward following 50% of all incongruent correct trials and 0% of the congruent correct trials, and the low reward on all other correct trials. This mapping was reversed in the group rewarded more on congruent trials.

- p. 8, Figure 2: I would add to the note that both meta-analyses have a different scaling on the x-axis. Furthermore, I would like to see a graph depicting not only difference scores, but the full design of this study, so that one can see whether the reduction in the congruency effect is due to congruent trials, incongruent trials, or both. Maybe, at least as another appendix ;-)

We thank you for this suggestion. We added the note to the Figure (2) and also replaced the bar plots in the main text with bar plots per condition (same for the individual data violin lots in the Supplementary Material).

Figure 2

Meta-analysis results and congruency effects across experiments

Note. A, C: Model estimates per experiment and estimated effect size across experiments for the accuracy (A) and RT (C) model. The x-axes of the accuracy and RT meta-analyses depict different scaling. μ = Mean effect size. B, D: Raw data plots showing the accuracy and RT congruency effects (interaction between group and congruency) including means and standard error bars. Plots including individual subject data are presented in S3. Significance stars refer to the LMM results. Accuracies are depicted as error rates for better visualisation.

Figure 7

Stimulus-specific and stimulus-unspecific reward effects in the second experiment half

Note. Top row: Effects displayed for stimuli of the reward condition rewarded with the high reward in the previous experiment half. Bottom row: Effects displayed for stimuli that were not previously rewarded with the high reward (bottom). Significance stars refer to the model results. Plots including individual subject data are presented in S5.

Violin plots in the Supplementary Material S3:

Results of main analyses (excluding the pilot experiment)

Note. Raw data plot showing the accuracy and RT congruency effect (interaction between group and congruency). The upper and lower hinges of the boxplot correspond to the first and third quartiles (the 25th and 75th percentiles). The solid points indicate the means in addition to the median. We used lines to project the congruency effects on the other group to ease comparison. These raincloud plots are adapted from Allen et al. (2019).

And S5:

Results of second half of second Experiment

Note. Raw data plot showing the accuracy and RT congruency effect (interaction between group and congruency) of the previously highly rewarded trials (A) and previously not highly rewarded items (B). The upper and lower hinges of the boxplot correspond to the first and third quartiles (the 25th and 75th percentiles). The solid connected points indicate the means in addition to the median.

- p. 8 ff., *Drift diffusion modelling: it should be explicitly mentioned that these are not preregistered, but exploratory analyses; furthermore, I was wondering if there was a reason to conduct both, the standard DDM and the DMC? And why are the methods on the diffusion modelling reported here and not in the methods section?*

We now added a sentence to clarify that the diffusion modelling was of exploratory nature. The reason we report them in the results section rather than the methods section is that sequential sampling models like DDMs, especially the DMC, might be less familiar to some readers. These models could require some background explanation, unlike linear mixed models which are more commonly known, for example. Due to their exploratory nature, we also fitted both versions of the model, the more known and widely applied standard DDM as well as the DMC as a newer adaptation of it, which is particularly designed for conflict tasks (see also reply to comment 3), p. 8, l. 155:

In order to better understand the cognitive processes (...) we turned to drift diffusion modelling (DDM) as an exploratory analysis.

- p. 14, Figure 5: on the left side of this figure, blue and orange are used to differentiate congruent and incongruent trials, while on the right side blue and orange are used to differentiate the two groups. I suggest to use a distinct set of colors for each factor. The note seems to explain only the left side of the figure without mentioning the panels on the right side.

We thank you for this suggestion and replaced the colours to consistently use a different pair of colours for reward group (blue and orange) and congruency condition (green and yellow). Pertaining to Figure 5, we decided to omit the model-estimated diffusion models altogether to avoid redundancy with the toy models presented in Figure 3 and to allow more space for plotting additional parameters (in response to comments of Reviewer #2). We also added an introduction of the right panel to the Figure note. We also added some additional explanation for the panels:

Figure 5

Modelling results

Note. Mean estimates of the Standard drift diffusion model (DDM; A) and Diffusion model of conflict (DMC, B) with boundary shifts plotted for each group separately. Bars reflect the standard error per group. The group rewarded more on incongruent trials seemed to increase their boundary more for incongruent trials than the group rewarded more on congruent trials. In fitting the models, we went with the common formalizations of the classic DDM, modelling boundary separation as the difference between boundaries and the DMC, modelling boundary as the relative difference of the lower and upper bound from the start of the evidence accumulation (here from 0).

Moreover, estimates are on different scales due to the DDM being fit based on seconds and the DMC based on milliseconds, based on Ulrich et al (2015).

- p. 16, second half: "... with the group rewarded more on incongruent trials showing larger switch costs than the group rewarded more on congruent trials" -> the second "incongruent" should read "congruent"

We thank you for pointing out this mistake and corrected it in the manuscript.

- pp. 17-20: the discussion feels overly short with respect to the number of analyses and results reported in this manuscript. To give one concrete example, I would have liked to see a discussion not only of the decision boundary, but also the drift rate effect.

We agree that the introduction and discussion were rather concise. Based on all reviewers' suggestions we now expanded on the introduction by including additional studies, e.g., the conflict monitoring framework and discussed the drift rate more thoroughly (see also response to the points of Reviewer #2, e.g., our optimality analyses including drift rate). p., 9, l. 180:

Changes in attentional control are commonly reflected in this [the drift rate] parameter (Bogacz et al., 2006; Ritz et al., 2022; Yee et al., 2022). For instance, changes in drift rate have been positively linked to reward incentives (Bustamante, Lieder, Musslick, Shenhav, & Cohen, 2021; Manohar et al., 2015), confidence judgments and reward satisfaction (Corlazzoli, Desender & Gevers, 2023), all of which may play an important role in this study.

And p. 11, l. 249:

The finding of similar adaptations in drift rate fits with current theories on cognitive control, as reinforcement or incentives have been linked to faster and more accurate responding (Bijleveld, Custers & Aarts, 2010; Krebs, Boehler & Woldorff, 2010; Manohar et al., 2015)(...)

And p. 17, l. 384:

Following a similar pattern as compared to the boundary shift, the group rewarded more on incongruent trials showed a smaller difference in drift rate for congruent and incongruent trials according to the DDM with shift, pointing at control adaptations not only in response caution but also to smaller differences in selective attention or

improved task focus between the two congruency conditions. This finding shows that control can be optimized in multivariate ways (Ritz et al., 2022), as also further supported by our optimality analyses. Assuming that participants' behaviour was driven by a reward maximization strategy, i.e., the maximization of rewards per time unit, both adjustments in drift rate as well as in boundary shift were adaptive strategies up to a certain point, after which boundary shifts ceased to be optimal.

- p. 21 ff., Methods: I found it hard to follow due to the simultaneous report of the three different experiments.

We appreciate this comment and now introduce each experiment separately in the methods section, pointing out for each how it differs with respect to the other experiments.

- p. 31, Appendix A: please consider a revision of Table A2. When reading the content of the table, I was very confused about the Frequency Low-High part, because there was no mentioning of congruency. I guess it makes sense if one takes the table note into account. But it is again very complicated to unpack the relevant information. I hope there is a more accessible way to provide this information.

We agree that this table was suboptimal for introducing our designs. We now removed this table from the Supplementary Material and instead introduced each design separately, with the respective details, in the methods section. We hope that this improved readability and made it easier to extract the relevant information.

Reply to Reviewer #2

[...]

I believe that the studies are well executed and that they address important questions.

However, there are few major weakness that I estimate to reduce the impact this paper will receive: i) a lack of theoretical clarity in the rationale of the study and in its motivation, ii) a tendency to oversell (and over-interpret) the results, iii) the adequacy and clarity of the computational modeling, iv) a relatively unclear/ineffective writing that doesn't help the

reader to understand the logic and the motivation of the study or its conclusion. Below I will try to explain each of these points with some examples.

We thank you for this encouraging review. In response to your comments, we now formulated some of our inferences more cautiously and provide more background for our theoretical framework by starting from the traditional distinction between learning processes and cognitive control. We also focus on the drift rate parameter in more detail and added optimality analyses to our paper. Below, we will point-by-point reply to the different comments and suggestions that were raised.

1. Rationale and motivation

As the authors state in their introduction, the research literature considers learning (reinforcement or supervised) and cognitive control as fundamentally different processes. This sets the stage to their present attempt to show that this distinction overlooks the ability to use reinforcement to adjust control.

IN a way, such a result is not novel, as in the early 1990s, Tzelgov et al (1992) and others, have shown that a mere manipulation of the frequency of neutral-words can modulate the magnitude of Stroop interference:

*Tzelgov, J., Henik, A., & Berger, J. (1992). Controlling Stroop effects by manipulating expectations for color words. *Memory & Cognition*, 20(6), 727–735.*

<https://doi.org/10.3758/BF03202722>

What the present study has that differs from the Tzelgov et al (and many others), is that here the type of stimuli does not vary in the amount of conflict, but the critical variable is the feedback that is contingent on stimulus congruency. It is not clear to me however, that this differences has anything to do with reinforcement learning. I will try to explain my lack of clarity below (which I expect to be shared by other readers). There are two main points:

We thank you for drawing our attention to of the work of Tzelgov et al. In the revised manuscript this study is now cited for the idea that control can be modulated through manipulating the proportion of incongruent trials. More specifically, we added a sentence to the introduction, prior to introducing our reinforcement learning hypothesis, p. 4, l. 61:

Likewise, other early work has shown that people tend to adopt different control strategies in response to different proportions of Stroop incongruent words, as

reflected in modulated congruency effects (Logan & Zbrodoff, 1979; Tzelgov et al., 1992, for reviews, see Braem et al., 2019; Bugg & Crump, 2012).

i) There is a good reason why learning is typically seen as orthogonal to control. As it was clearly explained in the classical model of Stroop (Cohen, Dunbar, McClelland, 1990), the problem of control is to overcome the effect of learning (which would result in a strong but irrelevant pathway dominating a weaker but relevant pathway). Unlike learning which takes multiple trials (i.e., practice), control can take place upon instruction (from a first trial), and allows us to respond in a task-relevant way (with little practice).

We agree with you that there are good reasons for conceptualizing learning and control as orthogonal as done in seminal connectionist models (Cohen, Dunbar, McClelland, 1990, *Psychological Review*). We also appreciate the difference between the “slower” learning of connection weights versus recruiting attentional control to manage conflicting pathways or to select weaker pathways at a given moment. Yet, we do not think that this framework invalidates our research question which we see as complementary to this account. That is, we aimed to test whether *control configurations* can be learned in and of themselves, as people are often not (or cannot be) instructed on the most optimal control setting. In other words, the current project tests the idea that cognitive control adjustments can also become the subject of learning (for related ideas, see Bustamante, Lieder, Musslick, Shenhav, & Cohen, 2021, *CABN*; Musslick, Saxe, Hoskin, Reichman, & Cohen, 2020, *PsyArXiv*).

ii) The authors distinguish between the learning of stimulus specific associations, and the learning of congruency, presenting their study as a case of the second. However, it is not clear to me that this situation has anything to do with learning in the sense of reinforcement (or supervised) learning algorithms. This is because learning algorithms are mediated by changes in synaptic connections and this study does little to show how a neural-network that carries out their task may use the feedback to make changes on the connections of the network to account for the outcomes. Instead, it seems easier to conceptualize of the congruency reward as a reminder given to subjects to pay attention to relevant task. In the case of congruent trials the reminder is not very effective (because both tasks point to the same response), however for incongruent stimuli the reward succeeds to remind the subjects that they should keep paying attention to the relevant task. I would thus expect that the results

would be very similar, if the congruency based reward would be replaced (after each trial or after every few trials) by a cue reminding the participants to pay attention to the relevant task.

We thank you for this suggestion and agree that we cannot rule out that rewards acted as a cue to pay attention to the relevant task rather than a reinforcement signal.

However, we believe that even if this were the case, this finding would support rather than defeat our claim that optimal control settings can be learned through the use of reinforcement signals (i.e., rewards). Second, if high rewards in our task acted as conscious reminders to pay more attention, we believe the resulting control modulations should not be congruency-specific (i.e., resulting in an upregulation of the boundary for incongruent trials specifically), but should be observed across both groups and congruency conditions. Instead, we believe our data show that people applied congruency-specific control strategies. Moreover, as noted also in our previous point, we argue that these control parameters, in our case the decision boundary, can be conceptualized as learned weights themselves (e.g., Simen, Cohen & Holmes, 2006, *Neural networks*). Taken together, we believe that the alternative explanation put forward would ultimately result in a similar computational model where the reward signal selectively reinforces either task-relevant weights or task-adaptive boundary settings. We now added the reference to the neural network to the Introduction and also refer to it in the Discussion, p. 3, l. 55:

However, recent theories have come to suggest that people may also rely on associative learning or reinforcement learning when it comes to the regulation of these arguably more abstract cognitive processes, i.e., the configurations or weights of control settings themselves (Abrahamse et al., 2016; Braem & Egner, 2018; Chiu & Egner, 2019; Doebel, 2020; Logan, 1988; Simen, Cohen & Holmes, 2006; Verbruggen et al., 2014).

And p. 16, l. 365:

The idea of boundary adjustments as a reward maximizing strategy has further gained attention in the design of neural networks, providing a plausible mechanism to be tested in actual performance (Simen, Cohen, & Holmes, 2006).

If this was the case, I don't think we would consider this to have anything to do with reinforcement learning, but rather to show that control can be adjusted by conscious

reminders. To conclude, without showing that the change in control can be obtained as a result of a learning algorithm based on the neural network that executes the task and is presented with the feedback that the subjects received, it is hard to really accept the motivation for considering this study to show anything qualitatively different from the previous demonstrations that control processes can be adapted (as in Tzelgov et al.)

We generally agree that it is hard to rule out the broader idea that these rewards acted as conscious reminders, but would also argue that reinforcement learning and activation of task sets due to retrieval cues might be two semantically different ways of describing the same process. That is, we believe that even if control was adjusted through task reminders this very act could still be accomplished through, or computationally explained by, reinforcement learning which is agnostic about whether task or reward cues are conscious. Similarly, models of cognitive control such as the conflict monitoring framework (Botvinick, Braver, Barch, Carter & Cohen, *Psychological Review*, 2001) can be argued to implement cognitive control (control unit activation) through reinforcement learning-like principles (i.e., delta rule) of previous conflict (e.g., Botvinick et al., 2001, equation 2).

That said, we agree with you that showing the change in control that we observed in our experiments in a neural network could be illustrative. However, we currently do not have the hands-on experience to set this up in a reasonable timeframe, and we discussed this issue with neural network experts in the field. They agreed that while such a model would surely be interesting, the main principles have already been demonstrated elsewhere (Simen, Cohen & Holmes, 2006), and similar arguments can be made through computationally simpler optimality analyses (see below), with the reasonable assumption that a sufficiently advanced neural network would be able to find this optimal point in parameter space. However, as we agree this discussion point is interesting, we now added a paragraph to the Discussion to address this, p. 16, l. 346:

It is important to note that additional cognitive processes may have contributed to the observed group differences in conflict processing. For instance, rewards, rather than having served as a reinforcement signal in the narrowest interpretation, may have acted as a form of conscious reminders to pay more attention to the task, particularly when following incongruent stimuli. Yet, even if the observed differences in control modulations were the mere consequence of participants' redirecting their attention,

the resulting lasting control modulations should neither be congruency-specific, nor inherently rule out a reinforcement learning account. As elaborated in the discussion, we believe that an encompassing learning perspective, in both cases, provides a more parsimonious explanation of cognitive control adaptations rather than conceptualizing attention and control as strictly orthogonal processes.

2) Over-interpretation of the results

To show a change in control (or selective attention) one needs to show that the subject is able to change focus from the irrelevant (but automatic task or dimension) to the relevant one. Such a change would typically correspond to consistent changes in both speed and accuracy (and not, as in this study, in a speed accuracy tradeoff). In other words, it should result in a change of the drift value (and not in the chance of the response boundary). In the later case (which is what the modelling seems to confirm) we have a modulation of the caution that subjects apply to some type of stimuli, but not an enhanced focus on the task-relevant dimension.

We appreciate the comment and agree that changes in control often occur through attentional weights acting on stimulus-response to task mappings, reflecting changes in the drift rate (Bustamante, Lieder, Musslick, Shenhav, & Cohen, 2021; Corlazzoli, Desender & Gevers, 2023; Manohar et al., 2015). Yet, a body of recent studies suggest that changes in cognitive control do not only affect the drift, but can also affect boundary settings (e.g., Frömer & Shenhav, 2022, *Neuroscience and Biobehavioral Reviews*; Ratcliff & Frank, 2012, *Neural Computation*) and that both (and other parameters) can be important for control adaptations, conceptualizing cognitive control as a multivariate optimization problem (Ritz, Leng & Shenhav, 2022, *Journal of Cognitive Neuroscience*). In a number of studies, a recurring finding has been that people can adjust their thresholds based on speed/accuracy instructions. Thus, threshold are often considered an important control parameter that can be readily adjusted in accordance with the current task goal (e.g., Forstmann, et al., 2008. *Proceedings of the National Academy of Sciences*; Ratcliff & Rouder, 1998, *Psychological science*). Similarly, Botvinick et al. (2001, *Psychological Review*) acknowledged in their conflict monitoring model that a helpful adaptation to the occurrence of errors, associated with high conflict, is to drive shifts in the trade-off

between speed and accuracy, inducing more accurate but slower responding (see their Simulations 2c).

Drift rate and threshold changes have distinct behavioural profiles (faster reaction times and better accuracy as a function of changes in drift and slower reaction times and better accuracy as a function of changes in boundary). As long as people can react in these distinct ways to instructions or incentive structures, we believe it is reasonable to assume that both parameters can be controlled. To show that both adaptations can be relevant in our study, we now conducted and implemented optimality analyses in which we simulated data based on different combinations of drift rates (0-0.5, step size = 0.033) and boundary settings (-0.05-0.1, step size = 0.01). Indeed, we find that both increases in drift rate as well as increases in boundary for the rewarded condition (congruent vs. incongruent trials) can lead to an increased discounted total reward rate, which we calculated as first proposed by Bogacz et al., 2006 (2006, *Psychological Review*), taking into account response times with fast responses being inherently rewarding as well:

$$\text{Discounted reward rate} = \frac{\text{Mean accumulated reward}}{RT + ITI}$$

Interestingly, these analyses also show that after a certain point it is no longer beneficial to increase the boundary, speaking for a cost-reward trade-off. See manuscript, p., 11, l. 249:

The finding of similar adaptations in drift rate fits with current theories on cognitive control, as reinforcement or incentives have been linked to faster and more accurate responding (Bijleveld, Custers & Aarts, 2010; Krebs, Boehler & Woldorff, 2010; Manohar et al., 2015).

To see if either differential adjustments to boundary shifts or drift rate on congruent or incongruent trials were adaptive in terms of optimizing performance, we also conducted additional optimality analyses. As we only modelled separate drift rates for each trial type in the DDM with shifts, these optimality analyses were restricted to this model. Specifically, we simulated 50000 trials for 256 agents with different combinations for each boundary shift (-0.05-0.1 in 16 steps of 0.01) and drift rate (0-0.5 in 16 steps of 0.033) while setting all other parameters to average estimates across both groups. Reward was discounted by dividing it through the reaction time plus the intertrial interval, to account for the rewarding effect of fast responses (based on Bogacz et al., 2006). Interestingly, these simulations showed that, indeed, it is

beneficial to upregulate one's boundary shift in response to the reward condition up to a certain point after which it ceases to be beneficial. Moreover, it is adaptive to increase one's drift rate for the reward condition (see Figure 6). While increasing one's boundary after a certain point ceases to be beneficial as the time cost increases without a significant benefit for additional reward, it is plausible that the drift rate is also constrained by biological factors that are not modelled here, such as attentional processing units.

Figure 6

Heat maps of analyses for drift diffusion model with boundary shift

Optimal boundary shift per group and congruency level

Optimal drift rate per group and congruency level

Note. Figure depicting mean discounted reward rates based on different combinations of A) boundary shifts and B) drift rates for each congruency condition per reward group. Scaling is set to five breaking points for rewards but differs between sub plots. Crosses depict estimated means and standard errors across participant estimates. The line segment for the optimal boundary shift plots depicts the difference to the initial boundary with coordinates (0, 0).

I could see that there may be a weaker sense of labeling such a change of stimulus specific caution as a (weaker) type of control, but this at least requires a clear discussion of the distinction as well as qualification of the results.

We thank you for this comment. We appreciate that changes in drift rate are important modulations of cognitive control, as acknowledged in previous response, yet we are hesitant to call changes in the boundary parameter weaker or less important.

Ultimately, what might be the more important change in parameter will often be very dependent on the task at hand. However, we fully agree with you that our manuscript benefit from a more clear discussion of this distinction, thanks to your comments, which we also hope the above-mentioned optimality analyses further contribute to. In a similar vein, we now made the distinction between different types of control adjustments clearer in our discussion, p. 17, l. 384:

Following a similar pattern as compared to the boundary shift, the group rewarded more on incongruent trials showed a smaller difference in drift rate for congruent and incongruent trials according to the DDM with shift, pointing at control adaptations not only in response caution but also to smaller differences in selective attention or improved task focus between the two congruency conditions. This finding shows that control can be optimized in multivariate ways (Ritz et al., 2022), as also further supported by our optimality analyses. Assuming that participants' behaviour was driven by a reward maximization strategy, i.e., the maximization of rewards per time unit, both adjustments in drift rate as well as in boundary shift were adaptive strategies up to a certain point, after which boundary shifts ceased to be optimal.

3) The adequacy and clarity of the computational modeling

For me a main weakness of the computational modeling is that it didn't rely on some accepted/standard computational modeling of conflict (such as, for example, Botvinick et al, 2001). I can understand applying DDM modeling as an attempt to simplify the task.

However, I am less familiar with the conflict version of the DDM, and I expect that many readers may be unfamiliar with it too.

Thus if one wishes to rely on it (rather than do the proper job of applying a mechanistic conflict model), one should, at least, explain this model in much more detail to the audience. As it reads now, there appear to be quite a number of unmotivated and unclear assumptions:

We appreciate the value of Botvinick's conflict monitoring model, yet we believe that it does not fully allow us to demonstrate our point on the learning of more *abstract* control parameter settings, as it is restricted to showing adaptations in drift rate as a function of changing weights between stimulus-response mappings. Moreover, the goal of our modelling was to better understand in which parameters learned control settings show, rather than modelling the learning process in itself, which is typically the goal of neural network or reinforcement learning models. We also believe that our choice of the diffusion model of conflict is well supported by the literature. For instance, since its publication in 2015, it has been adapted by different researchers in different disciplines (e.g., Evans & Servant, 2022, *Psychological Review*; Lin et al., 2020, *Psychological Science*). However, we agree that our introduction of this model was sparse and now elaborate on the assumptions in more detail, inspired by your more specific comments that follow. We will provide a point-by-point reply below.

i) why does the boundary shifts upwards? Most choice-RT studies seem to report, either fixed boundaries or boundaries that shift downwards (indicating a type of urgency)!

Collapsing bound models are commonly used to capture decisions made under time pressure, when participants feel a need to respond even in the absence of strong evidence. Indeed, this can reflect some type of urgency signal (e.g., Churchland et al., 2008; Tajima et al., 2016, 2019). We did not explicitly mention this hypothesis as our diffusion models were fitted agnostic to whether the boundary would shift up- or downwards. In other words, when fitting our models, a form of collapsing bounds was evaluated as well. However, we deemed it less likely to find evidence for such an urgency signal in our study as we used a conservative response deadline of 3000ms and only rewarded accuracy. The fact that we allowed it to shift upwards is novel (but has been hypothesized before, Frömer & Shenhav, 2022, *Neuroscience & Biobehavioral Reviews*), reflecting an adjustment in response to conflict that takes a while to detect. To address this comment, we now added a reference to collapsing bounds on p. 9, l. 197:

Several studies have suggested that modelling dynamic boundaries which change trial by trial can be helpful in capturing behaviour. For example, it has been shown that "collapsing" boundaries can help in modelling some urgency signal (e.g., Churchland et al., 2008; Tajima et al., 2016, 2019) or, alternatively, that people

strategically adjust their decision boundary in response to conflict on a trial by trial basis (...)

ii) *What is the functional form of the drift variable for the incongruent condition?*

iii) *The various variables associated with this conflict-DDM model need to be better explained...(peak-latency, peak-amplitude). At present those appear somewhat unclear and arbitrary.*

The drift for congruent and incongruent trials is approximated with a gamma function which can either be multiplied by a positive or negative coefficient, depending on congruency (being positive on congruent trials and negative on incongruent trials in the equation below). This inverted-U-shaped function starts at 0, approaches its maximum quickly and slowly declines back to 0 which makes it desirable for a drift that is first drawn to the incompatible boundary and then slowly drifts towards the compatible boundary on incongruent trials, or shows early facilitation towards the compatible boundary on congruent trials (Ulrich et al., 2015). The parameters of the gamma function are a shape parameter alpha (the larger it is, the more the probability density function shifts to the right). It has an amplitude beta and latency tau, determining how much and when it peaks.

$$v(t) = C * \beta * e^{-\frac{t}{\tau}} * \left[\frac{t * e}{(\alpha - 1) * \tau} \right]^{\alpha - 1} * \left[\frac{\alpha - 1}{t} - \frac{1}{\tau} \right] + v_c$$
$$v(t) = -C * \beta * e^{-\frac{t}{\tau}} * \left[\frac{t * e}{(\alpha - 1) * \tau} \right]^{\alpha - 1} * \left[\frac{\alpha - 1}{t} - \frac{1}{\tau} \right] + v_c$$

We now added more information on this form to the manuscript, p. 8, l. 163:

Extensions to the standard DDM (...) have further included the superimposition of conflicting activations of both controlled and automatic responses by means of modelling evidence accumulation with a gamma function (see S4 and Ulrich et al., 2015). This U-shaped function allows a response on incongruent trials to initially drift towards the boundary of the conflicting response before drifting towards the controlled correct response (...) The additional parameters comprise at least the peak latency and peak amplitude (describing when this peak occurs and how strong the activation is, see Figure 3). Another parameter which is often set to a fixed value is alpha describing how much the probability density function shifts to the right.

Supplementary Material S4:

The gamma function of the conflict model was fitted with a gamma function as described in Ulrich et al., (2015)

$$v(t) = C * \beta * e^{-\frac{t}{\tau}} * \left[\frac{t * e}{(\alpha - 1) * \tau} \right]^{\alpha-1} * \left[\frac{\alpha - 1}{t} - \frac{1}{\tau} \right] + v_c$$

$$v(t) = -C * \beta * e^{-\frac{t}{\tau}} * \left[\frac{t * e}{(\alpha - 1) * \tau} \right]^{\alpha-1} * \left[\frac{\alpha - 1}{t} - \frac{1}{\tau} \right] + v_c$$

with C being positive on congruent and negative on incongruent trials.

iv) Unclear/ineffective writing

I will give few examples (I will add questions/comments, IN CAPITALS) to explain difficulties to understand:

p. 3 (1st paragraph of the Intro):

To engage in goal-directed behaviour, we often rely on higher-order task representations. Actions that are shared across tasks are usually easy to implement, while task-specific actions require more reconfiguration. These necessary reconfigurations show as task-rule congruency effects, i.e., slower and more erroneous responding when dealing with conflicting task sets. It is generally assumed that these reconfigurations indicate that people exert “cognitive control”. In this study, we investigated whether people can learn adaptive control processes through reinforcement learning.

IT IS NOT CLEAR WHAT IS MEANT BY HIGHER ORDER REPRESENTATIONS (HIGHER COMPARED WITH WHAT?), NEITHER IT IS CLEAR WHAT THIS HAS TO DO WITH RECONFIGURATIONS AND WITH CONGRUENCY EFFECTS. PERHAPS AN EXAMPLE COULD HELP. I believe that the first paragraph needs to be as clear as possible (and free from jargon) to attract readers to keep reading.

We thank you for pointing out our use of jargon. To make these sentences clearer, we omitted the term “higher-order” which is not necessary in this sentence. We also added an example, p. 3, l. 38:

Actions that are shared across tasks are usually easy to implement, while task-specific actions require more reconfiguration. Imagine two tasks, doing the laundry and packing luggage. Some actions are shared between both tasks (folding clothes), while other actions are not (categorising by fabric only makes sense while doing the

laundry, not when packing). Now, when both tasks are carried out in close succession, shared actions will be easier to implement.

p. 3 (2nd paragraph).

Recent theories have come to suggest that people may also rely on associative learning or reinforcement learning when it comes to the regulation of these arguably more abstract cognitive processes (e.g., Abrahamse et al., 2016; Braem & Egner, 2018; Chiu & Egner, 2019; Doebel, 2020; Logan, 1988; Verbruggen et al., 2014). This idea is not new. In fact, although the seminal cognitive psychology work of Ulric Neisser similarly referred to these functions as “higher” mental processes (Neisser, 1967), it also suggested that humans should be able to acquire their own executive routines by learning through experience. Still, likely due to the dominant focus on these processes as being executive and under strategic control, studies on the (reinforcement) learning of these designated “higher” control functions are scarce.

THE TERMS MORE ABSTRACT AND HIGHER LEVEL ARE NOT CLEAR. WHAT EXACTLY IS ABSTRACT (IN EXECUTING, SAY, A STROOP TASK)? AND WHY IS HIGH LEVEL? (COMPARED WITH WHAT?) I SUPPOSE THAT WHAT THE AUTHORS INTEND HERE IS TO DISCUSS THE DISTINCTION BETWEEN LEARNING PROCESSES AND ATTENTIONAL CONTROL, BUT PLEASE SEE COMMENT ON THE RATIONALE & MOTIVATION, ABOVE...

We thank you for this comment. As stated in response to previous comments, we appreciate that the distinction between learning and attentional control has been made traditionally and that it has led to valuable insights. We now made sure to state this more explicitly. We now added the following paragraph to our introduction, p. 3, l. 51:

In connectionist models this [task rule] conflict arises from shared representations between tasks which create interference (M. M. Botvinick et al., 2001; Cohen et al., 1990; Musslick & Cohen, 2021). Thus, control is required to select task-relevant (conflicting) pathways and is traditionally viewed as orthogonal to learning, i.e., the slow building of these pathways and associated weights.

We also clarified what is meant with more abstract on p. 3, l. 55:

However, recent theories have come to suggest that (...) when it comes to the regulation of these arguably more abstract cognitive processes, i.e., the configurations or weights of control settings themselves.

And with “higher-order”, i.e., domain-general, p. 3, l. 58:

This idea is not new. In fact, although the seminal cognitive psychology work of Ulric Neisser similarly referred to these domain-general functions as “higher” mental processes (Neisser, 1967).

p. 16

Interestingly, we also found a significant Group by Transition interaction in the accuracy models, both in the set of previously highly rewarded (Est = -0.047, 95% CI [-0.092, -0.002], 98% [pd]), and not highly rewarded stimuli (Est = -0.061, 95% CI [-0.103, -0.019], 100% [pd]), with the group rewarded more on incongruent trials showing larger switch costs than the group rewarded more on incongruent trials.

group rewarded more on INCONGRUENT trials showing larger switch costs than the group rewarded more on INCONGRUENT trials.

THERE MUST BE SOME MISTAKE HERE...

We thank you for pointing out this mistake, as also noticed by Reviewer #1. We now corrected the second instance of “incongruent” to “congruent”.

In addition, I find that some of the Figures are not very clear.

For example, in Fig 5 (which is possibly the most important figure), the random trajectories are hard to see, so I suggest to eliminate them, so that one can focus on the model parameters.

We thank you for this comment and have decided to omit the random trajectories Figures from the manuscript.

It may help also to present not only the difference in parameters between the congruent and incongruent stimuli, but each of them separately. Also what is meant by "boundary separation"? separation between the two response boundaries or between the same response boundary in different conditions/groups?

We have now clarified what is meant with boundary separation in each model (see Figure caption) and added the according parameters to the Figure note:

Figure 5

Modelling results

Note. Mean estimates of the Standard drift diffusion model (DDM; A) and Diffusion model of conflict (DMC, B) with boundary shifts plotted for each group separately. Bars reflect the standard error per group. The group rewarded more on incongruent trials seemed to increase their boundary more for incongruent trials than the group rewarded more on congruent trials. In fitting the models, we went with the common formalizations of the classic DDM, modelling boundary separation as the difference

between boundaries and the DMC, modelling boundary as the relative difference of the lower and upper bound from the start of the evidence accumulation (here from 0). Moreover, estimates are on different scales due to the DDM being fit based on seconds and the DMC based on milliseconds, based on Ulrich et al (2015).

Finally, I found the mixed Method (All experiments grouped together) somewhat difficult to read, and making the understanding of the task as well as the motivation for the various experiments more difficult.

We thank you for raising this concern. We agree, and now present each experiment separately.

Reply to Reviewer #3

[...]

The paper is nicely written, and the data and modelling would be of interest to readers. I have a few comments.

It would be useful to share model code in OSF, particularly if this is the first effort to model and estimate shifts in decision boundaries.

We are glad to hear that you found our paper to be well-written and likely of interest to readers. We will make all of our code available on OSF upon publication.

*A recent relevant paper perhaps worth citing is Diao, L., Li, W., Chang, W., & Ma, Q. (2022). Reward Modulates Unconsciously Triggered Adaptive Control Processes. *I-Perception*, 13(1). <https://doi.org/10.1177/20416695211073819>*

Also potentially relevant:

*Wisniewski D., Reverberi C., Momennejad I., Kahnt T., Haynes J. D. (2015). The role of the parietal Cortex in the representation of task-reward associations. *Journal of Neuroscience*, 35(36), 12355–12365. <https://doi.org/10.1523/Jneurosci.4882-14.2015>*

*Hall-McMaster S., Muhle-Karbe P. S., Myers N. E., Stokes M. G. (2019). Reward boosts neural coding of task rules to optimise cognitive flexibility. *Journal of Neuroscience*, 39(43), 8549–8561. <https://doi.org/10.1523/JNEUROSCI.0631-19.2019>*

Etzel J. A., Cole M. W., Zacks J. M., Kay K. N., Braver T. S. (2015). Reward motivation enhances task coding in frontoparietal Cortex. *Cerebral Cortex*, 26(4), 1647–1659.

<https://doi.org/10.1093/cercor/bhu327>

We thank you for pointing us towards these relevant studies. We now acknowledge them in our introduction, p. 4, l. 67:

It is well established that reward affects control processes (for review see, e.g., Botvinick and Braver 2001). Also more recent studies showed, for example, that blocks of anticipated high vs. low reward can trigger adjustments of control (even in the absence of awareness, Diao et al., 2022), that reward prospect improves performance through improvements in task coding (Etzel et al., 2016; Hall-McMaster et al., 2019) and that people can be instructed to learn associations between task representations and rewards (Wisniewski et al., 2015). However, these studies do not test the idea that cognitive control can be tested through reinforcement learning, i.e., the learning from retrospective rewards.

Regarding the increase in boundary in high reward trials – the interpretation that the group rewarded more on incongruent trials increased their decision boundary more for incongruent trials than the group rewarded more on congruent trials is clear. The result that both groups upregulate decision boundaries on congruent trials is indeed surprising, and initially seems at odds with the congruency effect in RT. The discussion addresses how this can be interpreted for the group differences observed in accuracy, but can the authors make clearer whether these models and this interpretation can account for the congruency effects in RT and the unexpected greater congruency effect for the rewarded incongruent group? From my interpretation of Figure 5, it seems like the DDM with shift can but the DMC with shift cannot, is this correct? I should note that I have limited experience with drift diffusion models, but I imagine some readers will also, so some additional clarity here may be helpful.

We thank you for this comment and agree that the interpretation of the parameters is not straightforward. While it might seem counterintuitive, we reason that both the DDM and DMC with shift can account for the congruency effect in RT as it is also determined by the drift rate parameter which in both cases is more skewed towards the correct response for congruent trials. Regarding the group differences, because of the higher boundary for incongruent trials in the incongruent group, incongruent trials

in particular take longer to reach the boundary. This in turn leads to larger reaction times on incongruent trials, resulting in overall larger congruency effects in RT. We now provide some clarification for the congruency effect on p. 11, l. 237:

While a higher boundary for congruent trials might seem at odds with the general observation of congruency effects in RT, it is important to note that the RT is also determined by the drift rate, which was generally higher (i.e., faster) for congruent than incongruent trials.

For the data using previously reinforced stimuli, the authors note “we found a speeded processing of incongruent stimuli when they were rewarded more, rather than the seemingly slower processing reported in the main analyses across experiments without stimulus repetitions. This suggests that previously highly rewarded stimuli now acted as reward cues, which have been associated with invigorating effects that speed up responses.” – Could you explain why this type of reward is more likely to speed up responses than the reward in the experiments without stimulus repetitions?

We thank you for pointing out this was not unclear. We unpacked this sentence more and hope it is clearer now, p. 14, l. 298:

Perhaps, items previously highly rewarded now, when recognized as such, act as reward cues themselves, a possibility that did not exist in the design with no item repetitions. In line with our finding, such reward cues have been associated with invigorating effects that speed up (rather than slow down) responses (Bijleveld et al., 2010; Krebs et al., 2010).

P18. “It would be interesting for future research to formalize this idea as a credit assignment problem where policies at different levels of control can have different learning rates.” – I didn’t understand what this meant.

We now added an additional sentence for clarification, p. 16, l. 342:

Specifically, such studies could test whether participants are biased towards first assigning rewards to the stimulus-response level rather than their more domain-general control configurations, and whether this first level is also associated with higher learning rates.

20th Nov 23

Dear Ms Held,

Your manuscript titled "Reinforcement learning of adaptive control strategies" has now been seen by our reviewers, whose comments appear below. In light of their advice I am delighted to say that we are happy, in principle, to publish a suitably revised version in Communications Psychology under the open access CC BY license (Creative Commons Attribution v4.0 International License).

As you can see, Reviewer #2 had remaining hesitations about the contribution of your work. We did come to the decision that the advance it offers was sufficient for publication in Communications Psychology, but we ask you to make good use of this feedback to improve the Discussion and Limitation sections.

We therefore invite you to revise your paper one last time to address the remaining concerns of our reviewers and a list of editorial requests. At the same time we ask that you edit your manuscript to comply with our format requirements and to maximise the accessibility and therefore the impact of your work.

Please note that it may still be possible for your paper to be published before the end of 2023, but in order to do this we will need you to address these points as quickly as possible so that we can move forward with your paper.

EDITORIAL REQUESTS:

SUBMISSION INFORMATION:

OPEN ACCESS:

Communications Psychology is a fully open access journal. Articles are made freely accessible on publication under a [CC BY](http://creativecommons.org/licenses/by/4.0) license (Creative Commons Attribution 4.0 International License). This license allows maximum dissemination and re-use of open access materials and is preferred by many research funding bodies.

For further information about article processing charges, open access funding, and advice and

support from Nature Research, please visit <https://www.nature.com/commpsychol/article-processing-charges>

At acceptance, you will be provided with instructions for completing this CC BY license on behalf of all authors. This grants us the necessary permissions to publish your paper. Additionally, you will be asked to declare that all required third party permissions have been obtained, and to provide billing information in order to pay the article-processing charge (APC).

* **DATA AVAILABILITY:**

[link redacted]

Best regards,

Marika Schiffer on behalf of
Antonia Eisenkoeck

Antonia Eisenkoeck

Senior Editor
Communications Psychology

REVIEWERS' COMMENTS:

Reviewer #1 (Remarks to the Author):

The authors did a good job in revising the manuscript. The more traditional format of reporting the experiments much improved readability. The authors addressed all my previous comments to my satisfaction and I only have one very minor comment left:

The abbreviations DMC and DDM are introduced a second time in the text on p. 16.

In my opinion, the DDM modeling is very informative in itself. The authors make all their data openly available, so anyone interested in modeling this data in a different way is free to do so.

Reviewer #2 (Remarks to the Author):

I do find the manuscript somewhat more clear. I do appreciate the effort made to clarify the links with previous studies showing adaptations of response boundaries (Simen et al, 2006), and studies indicating within trial variation of the response boundary (REFs 53-55). This theme is probably the main contribution of the paper (to my judgment) so probably the introduction should focus on it more.

However, I am still not convinced about the contribution of the article to a interdisciplinary audience (to my view it could be more adequate for a specialized journal). Below I list few reasons for this.

I. The theory is still somewhat unclear and is not backed up by a solid mechanism.

i) While the authors provided an explanation to changes of the boundaries (upwards, within a trial) as a result of conflict, this only makes sense for incongruent stimuli. However, the results seem to show such upwards changes in congruent trials too, which should minimize conflict.

ii) The inference that results are due of reinforcement learning, rather than conscious reminders is not convincing. The authors say that in the latter case, there should be NO dependency on the congruence of the stimulus. However, the stimulus congruence obviously affects the task accuracy, and it makes sense that people make more effort (become more cautious) when they experience more errors (and when each error has more cost).

iii) In any case, the paper misses to deliver a computational implementation of the control regulation, based on a change of weights, without which the difference between reinforcement learning and other types of control remains unsubstantiated.

II. The paper still appears to oversell the novelty and contribution to the the literature for both the theoretical and the empirical components.

First, showing an adaptive change in the mechanism with practice on a task, is not a particularly

novel or surprising result, and moreover changes of response boundaries with practice, have been reported before (REF-14, also Bogacz, Holmes, Cohen 2010; QJEP, 63(5):863-91, not referred to).

Second, the paper does not provide an implementation of reinforcement learning that operates on a model that accounts for task performance. Instead it relies of very indirect models (such as conflict-DDM), which introduce external (not self generated, dynamic components).

Third the empirical results include quite a number of marginal effects (e.g, last line on p.6, with confidence-interval touching on zero).

Reply to Reviewer #1

The authors did a good job in revising the manuscript. The more traditional format of reporting the experiments much improved readability. The authors addressed all my previous comments to my satisfaction and I only have one very minor comment left:

The abbreviations DMC and DDM are introduced a second time in the text on p. 16.

We thank the Reviewer for the positive comments and corrected the abbreviations by only introducing them at the first occurrence.

Reply to Reviewer #2

I do find the manuscript somewhat more clear. I do appreciate the effort made to clarify the links with previous studies showing adaptations of response boundaries (Simen et al, 2006), and studies indicating within trial variation of the response boundary (REFs 53-55). This theme is probably the main contribution of the paper (to my judgment) so probably the introduction should focus on it more.

We thank the Reviewer for acknowledging our changes and improvement of the manuscript. We also thank them for pointing out the novel contribution of the effects found for boundary adjustments. Regarding their integration in the manuscript, we would like to remain transparent about the fact that they were not the main analyses but part of exploratory follow-up analyses. Our main, preregistered goal and finding was to document the behavioural reduction of congruency effects in the group rewarded more on incongruent trials, motivated by our new experimental design where each stimulus was presented only once. Therefore, in response to the Reviewer's comment, we now acknowledge these boundary findings more in the Discussion by outlining future directions (see responses below), but prefer to keep our introduction in line with our a priori and preregistered goal.

However, I am still not convinced about the contribution of the article to a interdisciplinary audience (to my view it could be more adequate for a specialized journal). Below I list few reasons for this.

I. The theory is still somewhat unclear and is not backed up by a solid mechanism.

i) While the authors provided an explanation to changes of the boundaries (upwards, within a trail) as a result of conflict, this only makes sense for incongruent stimuli. However, the results seem to show such upwards changes in congruent trials too, which should minimize conflict.

We agree with the reviewer that the specific result in our follow-up analyses of higher boundary shifts for congruent trials was unexpected, and our explanation was post-hoc. For these reasons, we were also cautious in our wording when discussing this modelling result in our Discussion section (“*The general increase for congruent trials across groups may seem surprising at first [...]. However, the observed pattern may be a result of the specific reinforcement schedule, and is compatible with previous research showing that people are only willing to invest effort if it pays off.⁶⁷[...]*”). We would like to reiterate that these specific findings were not part of our main, preregistered effect of interest, nor did they contradict our main findings. Our main effect of reduced congruency effects in the group rewarded more on incongruent trials was preregistered, in the expected direction, and planned with methodological rigor. We now add a disclaimer after the above-mentioned interpretation of the boundary shift effects on congruent trials:

Naturally, this interpretation should be taken with caution as it is derived post-hoc on the basis of exploratory follow-up analyses.

ii) The inference that results are due of reinforcement learning, rather than conscious reminders is not convincing. The authors say that in the latter case, there should be NO dependency on the congruence of the stimulus. However, the stimulus congruence obviously affects the task accuracy, and it makes sense that people make more effort (become more cautious) when they experience more errors (and when each error has more cost).

We agree with the Reviewer that the different reward schemes used in our design could have triggered different control settings in the two reward groups and believe there is a misunderstanding with regards to our previous response.

We concur that we have reason to expect group differences as a function of our congruency-specific reward manipulation. As the Reviewer described, there is a bigger cost of making errors for participants rewarded more on incongruent trials as they have to “work harder” to gain the same amount of rewards as compared to participants rewarded more on congruent trials. Therefore, we acknowledge that our congruency-specific manipulation, could result in general, “conscious” reminders. However, our main argument is that the observed control adjustments *following* such reminders should not be congruency specific. If the above account were the sole explanation of our findings, we expect performance adjustments (or increased caution or drift rate) to benefit both congruent and incongruent trials. However, we observed no main effects of group on accuracy, RT, nor on initial boundary separation. As we do, however, find congruency-specific adjustments depending on the reward condition, we view this as evidence in favour of our more retrospective learning account, while acknowledging that it is not mutually exclusive to the cueing account. Therefore, we believe that the sentence added in our previous response letter is still justified. We now rewrote it to make it clearer:

It is important to note that additional cognitive processes may have contributed to the observed group differences in conflict processing. For instance, rewards, rather than having served as a reinforcement signal in the narrowest interpretation, may have acted as a form of conscious reminders to pay more attention to the task, particularly when following incongruent stimuli. Yet, while offering a complementary explanation of our findings, this account cannot stand alone or inherently rule out a reinforcement learning account, because, if observed differences in control modulations were the mere consequence of participants' redirecting their attention, we would expect the resulting lasting control modulations to be global rather than congruency-specific.

iii) In any case, the paper misses to deliver a computational implementation of the control regulation, based on a change of weights, without which the difference between reinforcement learning and other types of control remains unsubstantiated.

We agree that our paper does not deliver such a computational implementation, which was also not our primary intention. We highly appreciate the value of such an

implementation though which we now suggest as a future direction in the Discussion section.

Limitations

To follow up on our main results, i.e., the reduction of congruency effects as a function of reinforcement learning, we used diffusion models as described above. These models were of exploratory nature to obtain better insights into which underlying cognitive mechanisms were affected by our reinforcement schedule. However, these models were not preregistered, and do not model the learning of these process parameters. Therefore, it would be interesting for future research to follow up on our findings by developing a more overarching computational model that can account for both congruency-specific reinforcement learning and control adaptations (linked to performance benefits), for example by combining an evidence accumulation module to a neural network, and assigning separate weights to congruency-specific boundary shift parameters that can become stronger with practice. Along those lines, it would be interesting to adapt neural networks such as by Simen and colleagues¹⁴ to capture adaptations in thresholds as a function of stimulus congruency.

II. The paper still appears to oversell the novelty and contribution to the the literature for both the theoretical and the empirical components.

First, showing an adaptive change in the mechanism with practice on a task, is not a particularly novel or surprising result, and moreover changes of response boundaries with practice, have been reported before (REF-14, also Bogacz, Holmes, Cohen 2010; QJEP, 63(5):863-91, not referred to).

We agree that changes in boundaries, e.g., as a function of reward rate have been reported previously and appreciate the additional reference which we added to our manuscript:

The idea of boundary (or speed-accuracy)⁶⁶ adjustments as a reward maximizing strategy has further gained attention in the design of neural networks, providing a plausible mechanism to be tested in actual performance¹⁴.

However, boundary adjustments in response to stimulus congruency have, to the best of our knowledge, not been shown empirically. Concerning our main behavioural effect of interest, i.e., the reduction in congruency effects as a function of reward, we only know of a few, very recent studies that have looked into this, i.e., Chen et al., 2021; Mittelstädt et al., 2023; Prével et al., 2021; Yang et al., 2022, referred to in our paper. However, while meaningfully advancing theory, these studies are limited in showing stimulus-independent control adjustments. Our study does not only eliminate this confound by using a unique set of stimuli, it also adds a direct comparison of what happens when stimuli do repeat (Experiment 2B).

Second, the paper does not provide an implementation of reinforcement learning that operates on a model that accounts for task performance. Instead it relies of very indirect models (such as conflict-DDM), which introduce external (not self generated, dynamic components).

We agree with the Reviewer, and added this idea as a future direction (see also reply to point above).

Limitations

To follow up on our main results, i.e., the reduction of congruency effects as a function of reinforcement learning, we used diffusion models as described above. These models were of exploratory nature to obtain better insights into which underlying cognitive mechanisms were affected by our reinforcement schedule. However, these models were not preregistered, and do not model the learning of these process parameters. Therefore, it would be interesting for future research to follow up on our findings by developing a more overarching computational model that can account for both congruency-specific reinforcement learning and control adaptations (linked to performance benefits), for example by combining an evidence accumulation module to

a neural network, and assigning separate weights to congruency-specific boundary shift parameters that can become stronger with practice. Along those lines, it would be interesting to adapt neural networks such as by Simen and colleagues 14 to capture adaptations in thresholds as a function of stimulus congruency.

Third the empirical results include quite a number of marginal effects (e.g, last line on p.6, with confidence-interval touching on zero).

We agree with the Reviewer that our main effect of interest is small, which we are very explicit about in the manuscript. In fact, we consider this size, and its comparison with the effect we observe when allowing for stimulus-reward learning, one of the main conclusions of our paper. Yet, while being small, it is in the preregistered direction and was tested with a more conservative two-sided, rather than one-sided test.